# Self-organized metabotyping of obese individuals identifies clusters responding differently to bariatric surgery

Dimitra Lappa[1]*, Abraham S. Meijnikman[2,3], Kimberly A. Krautkramer[4], Lisa M. Olsson[4], Ömrüm Aydin[2,3], Anne-Sophie Van Rijswijk[5], Yair I. Z. Acherman[5], Maurits L. De Brauw[5], Valentina Tremaroli[4], Louise E. Olofsson[4], Annika Lundqvist[4], Siv A. Hjorth[6], Boyang Ji[1], Victor E. A. Gerdes[2,3], Albert K. Groen[2,7], Thue W. Schwartz[6], Max Nieuwdorp[2,4], Fredrik Bäckhed[4,6,8], Jens Nielsen[1,9]*

1 Department of Biology and Biological Engineering, Systems and Synthetic Biology, Chalmers University of Technology, Gothenburg, Sweden, 2 Department of Internal and Vascular Medicine, Academic Medical Center, University of Amsterdam, Amsterdam, The Netherlands, 3 Department of Internal Medicine, Spaarne Gasthuis, Hoofddorp, The Netherlands, 4 Department of Molecular and Clinical Medicine, Wallenberg Laboratory, Sahlgrenska Academy, University of Gothenburg, Gothenburg, Sweden, 5 Department of Surgery, Spaarne Gasthuis, Hoofddorp, The Netherlands, 6 Faculty of Health Sciences, Novo Nordisk Foundation Center for Basic Metabolic Research, University of Copenhagen, Copenhagen, Denmark, 7 Department of Pediatrics, Laboratory of Metabolic Diseases, University of Groningen, UMCG, Groningen, The Netherlands, 8 Department of Clinical Physiology, Region Västra Götaland, Sahlgrenska University Hospital, Gothenburg, Sweden, 9 BioInnovation Institute, Copenhagen N, Denmark

* lappa@chalmers.se (DL); nielsenj@chalmers.se (JN)

**Data Availability Statement:** The metagenomics dataset supporting the conclusions of this article is available in the European Nucleotide Archive (ENA)

## Abstract

Weight loss through bariatric surgery is efficient for treatment or prevention of obesity related diseases such as type 2 diabetes and cardiovascular disease. Long term weight loss response does, however, vary among patients undergoing surgery. Thus, it is difficult to identify predictive markers while most obese individuals have one or more comorbidities. To overcome such challenges, an in-depth multiple omics analyses including fasting peripheral plasma metabolome, fecal metagenome as well as liver, jejunum, and adipose tissue transcriptome were performed for 106 individuals undergoing bariatric surgery. Machine leaning was applied to explore the metabolic differences in individuals and evaluate if metabolism-based patients' stratification is related to their weight loss responses to bariatric surgery. Using Self-Organizing Maps (SOMs) to analyze the plasma metabolome, we identified five distinct metabotypes, which were differentially enriched for KEGG pathways related to immune functions, fatty acid metabolism, protein-signaling, and obesity pathogenesis. The gut metagenome of the most heavily medicated metabotypes, treated simultaneously for multiple cardiometabolic comorbidities, was significantly enriched in *Prevotella* and *Lactobacillus* species. This unbiased stratification into SOM-defined metabotypes identified signatures for each metabolic phenotype and we found that the different metabotypes respond differently to bariatric surgery in terms of weight loss after 12 months. An integrative framework that utilizes SOMs and omics integration was developed for stratifying a heterogeneous bariatric surgery cohort. The multiple omics datasets described in this study reveal that the metabotypes are characterized by a concrete metabolic status and different

database under the accession number PRJEB47902, (https://www.ebi.ac.uk/ena/browser/view/PRJEB47902?show=reads). The transcriptomics dataset has been deposited in the European Genome-Phenome Archive (EGA) database under the accession number EGAS00001005704. The metabolomics dataset has been deposited in the MetaboLights repository. Due to patient privacy and ethical restrictions the clinical metadata, transcriptomics, metabolomics and datasets contain potentially sensitive information, imposed by GDPR. However, access to BARIA patient clinical data, transcriptomics and metabolomics requires additional permission that can be granted upon reasonable request to Gut-MMM consortium data access committee (info@backhedlab.se). In addition, scripts used for the processing and analysis of data can be provided upon request to Gut-MMM consortium data access committee (info@backhedlab.se).

**Funding:** The BARIA study is funded by the Novo Nordisk Foundation (NNF15OC0016798). The Novo Nordisk Foundation Center for Basic Metabolic Research is supported by an unconditional grant (NNF10CC1016515) from the Novo Nordisk Foundation to University of Copenhagen. The BARIA study is a Scandinavian-Dutch collaboration. Funding from Knut and Alice Wallenberg Foundation is also acknowledged. The computations and RNA Sequencing were enabled by resources provided by the Swedish National Infrastructure for Computing (SNIC) at C3SE (SNIC Computational Center of Chalmers University of Technology) partially funded by the Swedish Research Council through grant agreement no. 2018-05973. There was no additional external funding received for this study. The funders had no role in study design, data collection and analysis, decision to publish, or preparation of the manuscript.

**Competing interests:** The authors declare no conflict of interest.

responses in weight loss and adipose tissue reduction over time. Our study thus opens a path to enable patient stratification and hereby allow for improved clinical treatments.

## Introduction

Obesity is generally associated with several different comorbidities, with type 2 diabetes (T2D) and cardiovascular diseases among the most common, and cross interaction of metabolic responses from these co-morbidities makes it difficult to study metabolic alterations associated with obesity. Thus, there is an increasing interest to study heterogeneous diseases like obesity through the collection of multiple omics data from various cohorts [1–3]. Due to the heterogeneity of phenotypes within obese individuals it is, however, generally difficult to stratify cohorts into groups, e.g. individuals with or without the metabolic syndrome, that can be compared using traditional statistical methods when omics data are to be analyzed. The use of machine learning methods is therefore gaining more attention for understanding and deconvoluting multifactorial disease [4,5], in particular as it enables stratification of individuals in a given cohort, without a priori knowledge of clinical labels.

Obesity is a growing worldwide epidemic, with an estimated 1.9 billion adults being overweight and another 650 million being obese [6–8], and it is associated with increased risk of multiple comorbidities including T2D, hypertension, dyslipidemia, non-alcoholic fatty liver disease and various types of cancers [9,10]. Numerous clinical approaches have been proposed to model obesity and predict bariatric surgery outcomes, by using clinical parameters, artificial intelligence and comparing predefined patient groups [11–14]. Another clinical definition for describing individuals with multiple dysmetabolic morbidities, including obesity, is the metabolic syndrome, where obese individuals fulfill two out of these four criteria: 1) fasting glucose >100 mg/dl; 2) triacylglycerol > 150 mg/dl; 3) high-density lipoprotein(HDL) cholesterol <40 mg/dl for males and <50 mg/dl for females; 4) blood pressure above 130 systolic or 85 diastolic [15]. The multitude of co-existing metabolic perturbations may also mask associations between metabolic activities in different tissues, including the gut microbiota, hence posing a challenge in systematically studying obesity, its' implications and the outcome of surgical intervention with higher resolution. A systems biology approach on the other hand could offer detailed phenotypic profiling possibilities using omics analysis. Metabolomics has recently been proposed as an approach to better comprehend obesity and linked comorbidities [16–18] and identify optimal candidate groups for further interventions [19,20]. The gut metagenome is a contributing factor to the complexity of obesity [21–25], although it's causal role has yet to be established [26]. Recent studies have pinpointed that the production and regulation of metabolites of bacterial origin in humans, play an important role in metabolic diseases [24,27–30]. Given these interactions, there is a clear need to propose a systems biology framework to obesity population-based studies, to improve the identification of distinct sub-populations but also drive the development of personalized interventions [31].

With the objective of getting novel insight into how metabolism in different tissues varies in obese individuals and evaluate if grouping of patients according to metabolism is related to their weight loss response to bariatric surgery, we generated multiple omics datasets from 106 individuals undergoing bariatric surgery. Specifically, we wanted to evaluate if the heterogeneity of a bariatric surgery population can be stratified phenotypically using metabotyping, i.e. grouping according to the individuals fasting plasma metabolome, that captures the functional output of a complex multi-organ system, human hosts and their microbes rather than by

traditional clinical classifiers, e.g. the metabolic syndrome. For this we established a novel workflow that first utilizes metabolomics for unlabeled stratification of individuals with several comorbidities and different pharmacological treatment regimens. We then analyzed transcriptome data from liver, jejunum, mesenteric and subcutaneous adipose tissues along with shotgun metagenomic sequencing from fecal samples to produce a discriminatory multi-marker signature of underlying metabolic phenotypes within obesity. The framework is solely based on omics data types representative of various biological molecule classes (metabolome, transcriptome, metagenome) and machine learning, instead of comorbidities, medications, and disease-specific classifiers, thus making it suitable for studying multifactorial metabolic conditions, besides obesity.

## Methods

### Ethics approval and consent to participate

The study was performed in accordance with the Declaration of Helsinki and was approved by the Academic Medical Center Ethics Committee of the Amsterdam UMC. All participants provided written informed consent.

### BARIA cohort

The recruitment of participants was conducted from the BARIA [32] study with a total of 106 individuals included. The baseline characteristics of BARIA participants in the Self-Organizing Map (SOM)-defined metabotypes are described in Table 1.

Individuals underwent a complete metabolic work-up at the start of their bariatric surgery trajectory. Anthropometric measurements including height, weight and waist and hip circumference were taken. In addition, body fat percentage using bioelectrical impedance and blood pressure were measured. Fasting blood samples were used for the determination of hemoglobin, HbA1c, glucose, lipid profile, alanine aminotransferase, aspartate aminotransferase, insulin, and creatinine levels. Within three months before surgery, a 2-hour mixed meal tolerance test was performed to assess insulin resistance and investigate dynamic alterations in circulating metabolites. Within three months before surgery, a 2-hour mixed meal tolerance test (MMT) was performed to assess insulin resistance and investigate dynamic alterations in circulating metabolites. The MMT consisted of a compact 125ml drink (Nutricia®) containing in total 23.3 grams fat, 74.3 grams carbohydrates (of which 38.5 grams sugar) and 24.0 grams protein. The participants received this meal after fasting for a minimum of nine hours. Time point zero refers to the moment at which the participant had fully consumed the meal. Blood samples were drawn *via* an intravenous line at baseline, 10, 20, 30, 60, 90 and 120 minutes. All samples were stored at -80˚C until further processing.

### Metabolome analysis

EDTA plasma samples under fasting conditions were collected from 106 BARIA participants. Samples were shipped to METABOLON (Morisville, NC, USA) for performing analysis using ultra high-performance liquid chromatography coupled to tandem mass spectrometry (LC-MS/MS) untargeted metabolomics, as previously described [27]. The metabolomic counts obtained, underwent significant curation via metabolites' pre-filtering, imputation for subsets of metabolites' missing values and data normalization, in order to minimize the effect of artifacts in the downstream analysis. Out of 1345 metabolites measured by METABOLON, 652 metabolites were fully detected across all samples, 640 metabolites were partially detected across all samples, and 53 metabolites were not detected and therefore had a missing value.

**Table 1. BARIA cohort: 106 participants clinical metadata summary for SOM-defined clusters.**

| Clinical Metadata | SOM Cluster 1 | SOM Cluster 2 | SOM Cluster 3 | SOM Cluster 4 | SOM Cluster 5 | BARIA population |
|---|---|---|---|---|---|---|
| **Demographic** | | | | | | |
| Participants (%) | 17(16%) | 29(27.4%) | 25(23.6%) | 18(17%) | 17(16%) | 106(100%) |
| Female *(% Total Participants, % of SOM Cluster)* | 13(12.2%, 76.5%) | 25(23.6%, 86.2%) | 18(17%, 72%) | 14(13.2%, 77.8%) | 14(13.2%, 82.4%) | 84(79.2%) |
| Male *(% Total Participants, % of SOM Cluster)* | 4 (3.8%, 23.5%) | 4 (3.8%, 13.78%) | 7 (6.6%, 28%) | 4 (3.8%, 22.22%) | 3 (2.8%, 17.6%) | 22(20.8%) |
| **Anthropometric** | | | | | | |
| Age *(years)* | 48(29–60)* | 40(20–57)* | 53(26–64)* | 56(39–64)* | 44(22–62)* | 46(20–14) |
| BMI *(kg/m²)* | 39.5(34–45.4) | 38.2(32.9–60.9) | 39.8(33–57.5) | 38.3(33.8–47.1) | 39.8(34.7–46.4) | 39.42(32.9–70) |
| Waist circumference *(cm)* | 125.3 ± 12.6 | 122.6 ± 12.3 | 123.7 ± 11.5 | 125.8 ± 12.2 | 123 ± 9.9 | 84.3 ± 57.7 |
| Upper thigh circumference *(cm)* | 135(120–149) | 133(116–147) | 130(103–165) | 133(115–139) | 136(123–144) | 122.5(103–165) |
| Total Body Fat *(%)* | 53.6(41.6–64.7) | 54.1(31.7–94.9) | 51.8(39.3–104.8) | 56.5(40.6–78.9) | 57.6(44–64.5) | 51(31.7–104.8) |
| Fat Free Mass *(kg)* | 60.9(54.1–93.8) | 59.6(50.3–90.6) | 59.1(47.5–90.2) | 59.8(49.5–85.1) | 60.8(54–83.5) | 58.9(47.5–93.8) |
| Systolic blood Pressure *(mmHg)* | 131.5(116–156) | 132(102–155) | 133(108–161) | 136(115–193) | 135(115–157) | 132.5(102–193) |
| Diastolic blood Pressure *(mmHg)* | 84.5(59–91) | 81(54–99) | 82(67–105) | 80(45–121) | 82(65–94) | 81(45–121) |
| **Clinical lab values** | | | | | | |
| Fasting glucose *(mmol/l)* | 5.8(4.8–11.4) | 5.9(4.6–14.8) | 5.7(5–13.8) | 5.8(4.6–6.8) | 5.6(4.5–8.7) | 5.8(4.5–14.8) |
| HbA1c *(mmol/mol)* | 5.7(5.3–9.1) | 5.7(4.6–9.8) | 5.6(5–9.3) | 5.8(5.2–6.9) | 5.5(5.2–8.3) | 5.7(4.6–9.8) |
| HOMA-IR | 1.7(0.6–3.4) | 1.6(0.5–6.9) | 2.2(0.5–4.7) | 1.3(0.8–4.8) | 1.5(0.8–4.8) | 1.6(0.6–6.9) |
| HOMA2-β | 108.7(38.3–183.2) | 87.9(29.1–227.8) | 112(52.7–226.2) | 92.1(52.4–357.8) | 104.2(50.8–185.5) | 93.5(29.1–357.8) |
| Total Cholesterol *(mmol/l)* | 5.4 ± 1.1 | 4.6 ± 1 | 4.9 ± 1.1 | 5.3 ± 1.2 | 4.3 ± 0.9 | 4.9 ± 1.1 |
| Triglycerides *(mmol/l)* | 1.5(0.8–3.5) | 1.3(0.6–5.8) | 1.4(0.8–6) | 1.4(0.8–5.9) | 1.2(0.6–1.9) | 1.4(0.6–6) |
| HDL Cholesterol *(mmol/l)* | 1.2(0.8–1.8)* | 1.1(0.6–1.9)* | 1.1(0.7–2.5)* | 1.2(0.7–2.1)* | 1.2(1–2.7)* | 1.6(0.2–2.7) |
| LDL Cholesterol *(mmol/l)* | 3.6 ± 1.1 | 2.9 ± 0.9 | 3.6 ± 0.9 | 3.4 ± 1.7 | 2.6 ± 0.8 | 3 ±1.1 |
| Creatinine *(μmol/l)* | 68(55–96) | 63(46–83) | 66(47–112) | 75(56–172) | 65(58–99) | 66(46–172) |
| Glomerular Filtration Rate *(kl/1.73m2)* | 85(70–91)* | 90(71–91)* | 86(62–91)* | 78(26–90)* | 89(66–91)* | 88.5(26–91) |

Baseline characteristics of the 106 BARIA participants included in the study. Data is expressed as mean ± standard deviation. Categorical variables are presented as numbers and percentages. Non-normally distributed variables are presented as median with interquartile range. For comparison among metabotypes Kruskal-Wallis test (extended Mann-Whitney *U*-test for multiple groups) was used.

'*' denotes differentially significant variables among the five metabotypes clusters ($P < 0.05$). BMI: Body Mass Index, HbA1c: Hemoglobin A1c, HOMA-IR: Homeostatic Model Assessment of Insulin Resistance, HOMA-β: Homeostatic Model Assessment of beta-cell function, LDL: Low-Density Lipoprotein, HDL: High-Density Lipoprotein.

The mean number of detected peaks (absolute abundance) for the fully detected metabolites in the BARIA cohort was 52 583 199. Whereas the mean absolute abundance for fully detected metabolites was 1 817 049. Metabolomics prefiltering and imputation were performed by utilizing a variation of the Perseus platform [33]. Essentially, data has been pre-filtered so as to have a maximum of 25% missing values for a metabolite across all samples. This was followed by a log transformation of all the measured metabolites' raw intensities across the entire dataset. Then, we calculated the total data mean and standard deviation (by omitting missing values). Taking into account that the metabolite intensities distribution is approximately following normality, we chose a small distribution 2.5 standard deviations away from the original data mean towards the left tail of the original data distribution, with 0.5 standard deviations

width. This new shrunken range corresponds to the actual lowest level of detection by the spectrometer. Here by drawing random values from this mini distribution, we filled the missing prefiltered data of choice.

Normalization was conducted to the total signal for each sample, since each sample is a separate injection on the mass spectrometer. Effective control for changes in sample matrix affects ionization efficiency, hence there can be inevitable differences in how much each sample is loaded onto the column with each injection, etc. Therefore, we summed up the total ion intensity (i.e. total signal) for each of the samples and identified the sample with the lowest total signal. After this we could proceed to calculating the correction factor for each sample by dividing the total signal with the lowest total signal,

$CorrectionFactor_i = \frac{Total\ signal\ for\ each\ individual\ sample_i}{Lowest\ total\ signal\ intensity}$. The next step is to divide each individual metabolite within a sample with the respective $CorrectionFactor_i$ After imputation and normalization, we obtained 986 metabolites. All the calculations for imputing and normalizing the metabolomics dataset have been conducted with MATLAB_R2018b and the standard built-in packages.

Differential analysis was conducted among the five SOM-defined Clusters in R (version 3.6.3) and RStudio (version 1.2.5033). Statistical analysis has been performed for fasting peripheral plasma with two methods: ANOVA(Analysis of Variance) and Kruskal Wallis test, with the use of HybridMTest package [34]. HybridMTest performs hybrid multiple hypothesis testing using empirical Bayes probability. The significance level and cut-off used for the dataset of fasting peripheral plasma was $P<0.05$ and was applied to metabolites that were significantly differential with both ANOVA and Kruskal Wallis methods.

## Clustering metabolome profiles with self- organizing maps

The fasting peripheral metabolomics were then input to the SOM toolbox [35] algorithmic setup in MATLAB_R2014b. SOMs conducted unsupervised competitive learning and produced low-dimensionality visualizations by employing vector quantization [36,37], a topology preserving projection. SOMs are essentially networks consisting of neurons in a lower dimensional space than the initial dataset, visually represented in a 2-dimensional grid. Each neuron has d-dimensions, equal to the number of features of the dataset and acts as a weight vector. During the SOM training phase, the weight vectors are gradually shifted in each iteration of learning, and the map gradually gets organized, so that neurons that are neighbors on the grid get similar weight vectors throughout the iterative training.

In our analysis, SOM took as input a set of prototype vectors representing the data. Every data item, here BARIA subject's fasting metabolome, was mapped into one point (node or neuron) in the map [38]. Mapping took place throughout the training phase of the SOM. The number of nodes was calculated internally by a heuristic formula, given the number of input vectors and their dimensionality, as $\sim 5 * \sqrt[2]{N}$, where $N$ is the number of data items and the number was slightly altered in order to fit hexagonal (instead of rectangular) nodes. The training method deployed in our study was batch training, where instead of taking each input vector separately and assigning a weight vector, the dataset was given to the SOM as a whole and the new weight vectors are weighted averages of the data vectors. In order to assign the prototype vector to the node, the Euclidean distances among prototype vectors and each neuron were calculated and set as the metric for the similarity measure. The "winner" node in the grid, was the one with the smallest Euclidean distance from the input vector. Once the assignment was complete, then the weights of the prototype vector along with the weights of the subset of its spatial neighbors in the array, got updated [39,40]. This entailed that all these local re-arrangements would be propagated along the grid, during the training epochs. As a result,

after learning, more similar data items would be associated with nodes that are closer in the array, whereas less similar items would be situated gradually farther away in the array.

When having a very large number of SOM nodes, one cannot easily quantify the results, hence the need for further grouping with a partitive approach. The resulting map was then subjected into $k-means$ clustering, as a built-in function of the SOM toolbox, for obtaining a recommended partition of map nodes. An open question in this case was the number of clusters, since $k-means$ in general takes this as a predefined parameter. Since $k-means$ is sensitive to initialization, we ran a cross validation simulation for 100 times for each $k$ (starting from $\sim 5 * \sqrt[2]{N}$, which corresponds to the number of nodes of the neural network to 1 with step of -1) for each with different random initializations. The best partitioning for each number of clusters was selected using error criterion and the minimization of the Davies-Bouldin cluster validity index [41]. Davies-Bouldin index is a metric of the ratio of the within cluster scatter, to the between cluster separation. The index's value is essentially the average similarity between each cluster and its most similar one, averaged over all the clusters. This implies that the best clustering scheme minimizes the Davies-Bouldin index. Eventually, when all the iteration for the potential values of $k$ were concluded, the minimum overall Davies-Bouldin index was chosen, which resulted in the recommended partition of five clusters.

## Transcriptome analysis

Biopsies from liver (106 samples), jejunum (105 samples), mesenteric adipose fat (104 samples) and subcutaneous adipose fat (105 samples) were collected at the time of the bariatric surgery, as previously described [32]. RNA was extracted from biopsies using TriPure Isolation Reagent (Roche, Basel, Switzerland) and Lysing Matrix D, 2 mL tubes (MP Biomedical, Irvine, CA, USAs) in a FastPrep®-24 Instrument (MP Biomedical, Irvine, CA, USAs) with homogenization for 20 seconds at 4.0 m/sec, with repeated bursts until no tissue was visible; homogenates were kept on ice for 5 minutes between homogenization bursts if multiple cycles were needed. RNA was purified with chloroform (Merck, Darmstadt, Germany) in phase lock gel tubes (5PRIME) with centrifugations at 4˚C, and further purified and concentrated using the RNeasy MinElute kit (Qiagen, Venlo, The Netherlands). The quality of RNA was analysed on a BioAnalyzer instrument (Agilent), with quantification on Nanodrop (Thermo Fisher Scientific, Waltham, MA, USA). Due to degradation of the RNA, libraries for RNAseq sequencing were prepared by rRNA depletion; library preparation and sequencing were performed at Novogene (Nanjing, China) on an HiSeq instrument (Illumina Inc., San Diego, CA, USA) with 150 bp paired-end reads and 10G data/sample. The average read count per sample from liver and jejunum tissues were 42 ± 15 million. For mesenteric and subcutaneous fat, the average read count per sample were 43.2 ± 20 million.

The extracted fastq files were analyzed with nf-core/rnaseq [42], a bioinformatics analysis pipeline used for RNA sequencing data. The workflow processed raw data from FastQ inputs (FastQC, TrimGalore!), aligned the reads (STAR) with *Homo sapiens* GRCh38 as reference genome, generates gene counts (featureCounts, StringTie) and performed extensive quality-control on the results (RSeqQC, dupRadar, Preseq, edgeR, multiQC). The pipeline was built using Nextflow.

Differential gene expression analysis for five SOM defined cluster participants has been performed for liver, jejunum, subcutaneous adipose and mesenteric adipose tissues, respectively, in R (version 3.6.3) and RStudio (version 1.2.5033) with DESeq2 [43] package. The statistical analysis method for calculating differential expression rates was the LRT test (log-ratio test). After FDR correction (FDR 5%) with multiple hypothesis testing with IHW [44] package, we analyzed genes with *P<0.05* by DEGreport's [45] degPatterns function, so as to identify

subgroups of co-expressed genes across the SOM clusters and assign a *z score* to each metabotype. For these differentially significant co-expressed genes we performed gene enrichment with Enrichr platform [46] using KEGG(Kyoto encyclopedia of genes and genomes) metabolic pathways [47]. Adjustment for the confounding factors of age and gender was conducted via the built-in function of DESeq2.

## Microbiome analysis

Fecal samples from 106 participants (108 fecal samples due to having two samples from two participants) were collected on the day of surgery and immediately frozen at -80C. Total fecal genomic DNA was extracted from 100 mg of feces using a modification of the IHMS DNA extraction protocol Q [48]. Briefly, fecal samples were extracted in Lysing Matrix E tubes (MP Biomedical, Irvine, CA, USA) containing ASL buffer (Qiagen), and lysis of cells was obtained, after homogenization by vortexing for 2 minutes, by two cycles of heating at 90˚C for 10 minutes followed by three bursts of bead beating at 5.5 m/sec for 60 seconds in a FastPrep®-24 Instrument (MP Biomedical, Irvine, CA, USAs). After each bead-beating burst, samples were placed on ice for 5 minutes. The supernatants containing fecal DNA were collected after the two cycles by centrifugation at 4˚C. Supernatants from the two centrifugations steps were pooled and a 600 μL aliquot from each sample was purified using the QIAamp DNA Mini kit (Qiagen, Venlo, The Netherlands) in the QIAcube (Qiagen, Venlo, The Netherlands) instrument using the procedure for human DNA analysis. Samples were eluted in 200 μL of AE buffer (10 mmol/L Tris·Cl; 0.5 mmol/L EDTA; pH 9.0). Libraries for shotgun metagenomic sequencing were prepared using a PCR-free method; library preparation and sequencing were performed at Novogene (Nanjing, China) on an HiSeq instrument (Illumina Inc., San Diego, CA, USA) with 150 bp paired-end reads and 6G data/sample.

MEDUSA [49] pipeline was used for pre-processing of raw shotgun metagenomics sequence data. MEDUSA is an integrated pipeline for analysis of short metagenomic reads, which maps reads to reference databases, combines output from several sequencing runs and manipulates tables of read counts. The input number of total reads from the metagenome analysis were on average 23.4 ± 2.2 million reads per sample and the total aligned reads 16.6 ± 1.8 million reads per sample. The sequencing runs had high quality with almost 98% of the reads passing the quality cut-off (~(20 million reads per sample). Out of the high-quality reads, on average 0.04% aligned to the human genome, although the data had been cleaned for human reads. Out of the high quality non-human reads, 78.4% aligned to MEDUSA's software gene catalogue. Quality filtered reads were mapped to a genome catalogue and gene catalogue using Bowtie2 [50]. Statistical analysis was performed in R (version 3.6.3) and RStudio (version 1.2.5033) on rarefied count, (20 million reads per sample). The taxon ids were input to taxize [51] package, so as to get full taxonomic information and ranking for the species. This dataset was input to DESeq2 [43] and phyloseq [52] packages for conducting downstream differential statistical analysis. Similar to the BARIA transcriptomics counts, log normalization has been conducted based on gene counts geometric distribution. Statistical analysis test for calculating differential expression rates was LRT. The IHW package, as part of DESeq2 suite, is utilized for multiple hypothesis testing and adjusting the respective P values, with alpha significance threshold set at *P<0.05 and FDR at 5%*. Adjustment for the confounding factors of age and gender was conducted via the built-in function of DESeq2.

## DIABLO correlation analysis and biomarkers minimal signature

DIABLO [53] stands for Data Integration Analysis for Biomarker discovery using Latent cOmponents and performs supervised multi-omics data integration, by maximizing the correlation

between co-expressed elements in the input datasets. DIABLO algorithm extends sparce Generalized Canonical Correlation Analysis [54] and by expanding the Partial Least Squares (PLS) regression, used singular value decomposition for dimensionality reduction and selected co-expressed (correlated) variables that could explain the categorical outcome of interest, in our case the five SOM-derived metabotypes. DIABLO analysis was conducted in R (version 3.6.3) and RStudio (version 1.2.5033) through the package of mixOmics [55] (version6.10.9). DIA-BLO output a set of latent variables (components) based on the dimensionality and the importance of the input datasets. All the datasets in this study carried the same weight, hence the DIABLO dataset matrix initialization design parameter was diagonal. The original input was 289 metabolites, 119 microbial species and 776 genes, all the differentially identified components from the omics datasets. This chosen number of components could extract sufficient information to discriminate all SOM-defined metabotypes. Then, a set of coefficients was attributed to each variable, that indicated the importance of each variable in DIABLOMulti-omics Datasets. The goal was to have maximization of the covariance between a linear combination of the variables from each input dataset and each categorical outcome. The algorithm was optimized with a 10-fold validation over 10 training epochs. After tuning these two hyper-parameters (number of variables from each dataset, choice of variables that maximize co-variance), DIABLO produced as output a minimal signature of total 113 markers that distinguish the given metabotypes.

## Results

### Metabolomics based stratification of bariatric surgery population via SOM

To create a multi-omics profile of obesity, a total of 106 individuals from the BARIA [32] bariatric surgery cohort were recruited. The multiple omics analysis included metabolomics on fasting peripheral blood samples, and we employed this dataset for stratification of the heterogenous group of individuals, independent of traditional clinical indexes, such as Body Mass Index (BMI), hypercholesterolemia, hypertension and treatment for T2D. To enable stratification based on the metabolomics data we built an unsupervised artificial neural network that could group individuals based only on the similarity of their metabolome, a SOM. The SOM [36] evaluated metabolomic similarity by calculating the Euclidean distances between complete metabolomic profiles, and projected BARIA individuals with high inter-group similarity onto a "map" of lower dimensionality compared to that of the initial dataset. The SOM was trained with 106 prototype vectors, where each prototype vector corresponded to a BARIA participant's peripheral plasma metabolite profile, consisting of 986 metabolites (see **Methods**). Iterative training of the SOM resulted in a map of 48 nodes, all projected onto a hexagonal grid (Fig 1A).

These 48 nodes considerably reduced the dimensionality and further within-cluster variance was minimized using $K-means$ [56], which preserved metabolomic distances and identified centroids of core metabolomes.

### SOM and k-means clustering reveal five distinct metabotypes

Clustering the SOM with $k-means$ identified five clusters (metabotypes), each with different features (Fig 1A, Table 1), including unique distributions of comorbidities and medication usage. Polypharmacy is a notable characteristic within this study population, including use of medication for T2D (n = 20), hypertension (n = 30), hypercholesterolemia (n = 42) and gastro-esophageal reflux disease (GERD, n = 16). Medication usage was distributed across the five metabotypes and is shown in Fig 1B. Clusters 1 and 3 include most individuals simultaneously treated for hypertension and high cholesterol (four and five individuals respectively), whereas

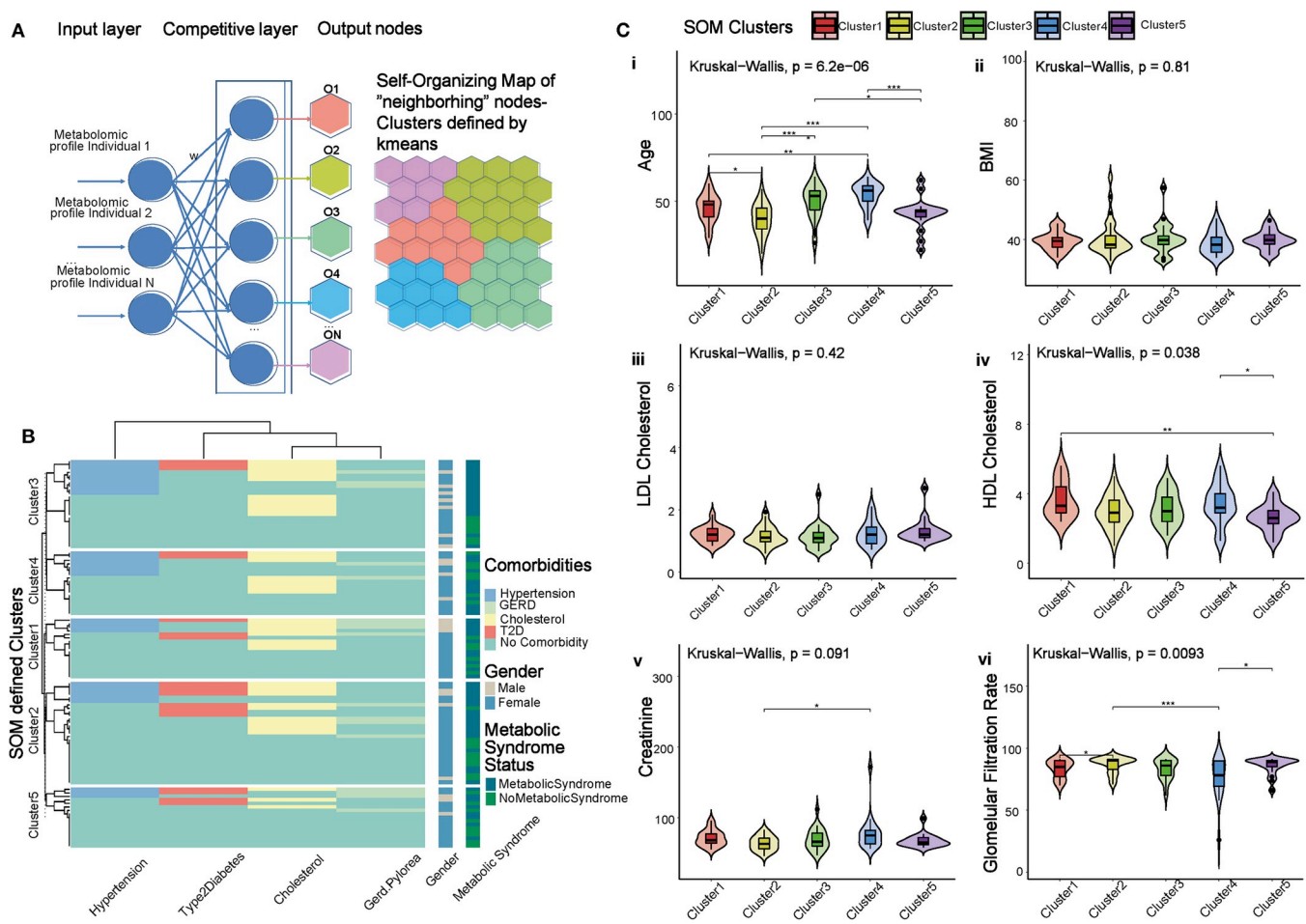

**Fig 1. Self-organizing maps reveal five distinct metabotypes within BARIA cohort. (A)** Architecture of a competitive artificial neural network. Each individual's complete metabolomic profile is assigned a weight. The weights are in turn assigned to neurons in the competitive layer of the neural network. In the competitive layer, SOM algorithm calculates the similarity metric (here Euclidean distance) between each metabolomic profile and the neurons and then updates the weights. After training, the network assigns the individual's metabolomic profile to the "winner" output node, the node that is essentially more similar to the input metabolomic profile. Once this step is complete, all the nodes are comprising the SOM. Finally, all the nodes of the SOM are subjected to k-means clustering resulting in the partitioned topology, the metabotypes (SOM & k-means defined clusters). **(B)** Clustergram of hierarchical cluster analysis depicting the distribution of medically treated cardiometabolic comorbidities of the individuals in each of the metabotypes (SOM & k-means defined clusters). The treated comorbidities are: hypertension, T2D, GERD and cholesterol. In parallel columns are the gender and metabolic syndrome status of each individual, respectively. **(C)** Clinical variables associated with obesity and their statistical significance across the metabotypes (SOM & k-means defined clusters): age **(C. i)**, BMI **(C. ii)**, HDL cholesterol **(C. iii)**, LDL cholesterol **(C. iv)**, creatinine and **(C. v)**, glomerular filtration rate **(c. vi)**; statistical significance among metabotypes is calculated with Kruskal-Wallis test; the symbols indicating significance among metabotypes are '*': P< = 0.05, '**': P< = 0.01, '***': P< = 0.001.

cluster 2 includes individuals co-treated for hypertension, high cholesterol and T2D (four individuals). Nevertheless, the distribution of overlapping treated cardiometabolic comorbidities is quite uniform among clusters and not skewed towards a particular metabotype. In order to assess the effect of missing values and data imputation in SOM clustering, a separate mapping analysis was conducted by using the unimputed metabolomics dataset. The final map clustering did not diverge from the original prediction. Hence, the ability of the SOM to assign similar items on the same node was not affected by the imputation of a minimal set of missing metabolite values. We next assessed the biometric features of each metabotype, by performing differential analysis on the clinical variables available. BMI, body fat, and waist circumference did not significantly differ between clusters, however age, HDL cholesterol and glomerular filtration rate varied between clusters (Fig 1C). Given that all BARIA participants are affected by

severe obesity, the stratification based on their SOM metabolomic profile reveals that BMI and treatment of cardiometabolic comorbidities are not the clinical features more accurately describing and differentiating the metabotypes, but age, cholesterol and markers associated with kidney function are important features. We also evaluated if there is any association within the clusters and individuals having the metabolic syndrome and found that there was no such association (Fig 1B). Furthermore, if we grouped the individuals according to having or not-having the metabolic syndrome, we also found no separation according to age or other clinical parameters besides those defining the metabolic syndrome (S1 Fig).

## Metabolomic profiles characterized by lipid and amino acid metabolites

Following stratification of the individuals into the five metabotypes we performed differential analysis of the metabolome for the five different metabotypes. Statistical analysis revealed 289 differentially significant metabolites. In comparison we only identified 3 differentially significant metabolites when the cohort was grouped according to presence of the metabolic syndrome or no (S2 Fig), which shows that driving grouping of the cohort based on the metabolomics data enables more detailed insight into metabolic differences among the individuals. KEGG pathway analysis revealed that the most highly enriched metabolite classes among the 289 metabolites were lipids, amino acids and xenobiotics, followed by cofactors and vitamins, nucleotides, carbohydrates, peptides, energy and partially characterized molecules. Clusters 2 and 3 exhibited the highest relative abundance of differentially significant metabolites, mainly lipids and amino acids (Fig 2A).

Among the enriched KEGG metabolic pathways that had the highest number of differentially significant metabolites were fatty acids (Fig 2B), specifically 19 lysophospholipids, 16 dicarboxylate fatty acids, 14 sphingomyelins and 12 phosphatidylcholines. The amino acid metabolic pathways with the most significant metabolites were arginine and proline metabolism with 11 compounds, tyrosine metabolism with 8 metabolites, methionine, cysteine SAM and taurine metabolism with 8 metabolites, too, while branched-chain amino acid metabolism for isoleucine and valine had 7 metabolites. The top 20 differentially abundant metabolites are a mixture of lipids, partially characterized molecules, peptides and amino acids, and some of them, despite being the end product of endogenous ketogenesis produced by the liver, also carry the potential of being the result of gut microbial metabolism, such as 3-hydroxybutyrate and acetoacetate (Fig 2C). Our analysis identified lipid metabolites (especially lysophospholipids and sphingomyelins) and amino acid metabolites (urea, arginine and proline metabolism) being significantly altered among the clusters.

## Hepatic and adipose tissue transcriptomes enriched for immune, amino acid and lipid metabolism functions

To better understand the relationship between metabolite levels and gene expression, we next sequenced RNA extracted from biopsies taken during bariatric surgery from liver, jejunum, mesenteric adipose tissue and subcutaneous adipose tissue. We identified differentially expressed genes between clusters and conducted gene set enrichment analysis. This analysis revealed 682 hepatic genes differentially expressed across the five metabotypes. In contrast, only four genes were differentially expressed in jejunum, whereas 45 and 49 genes were differentially expressed in mesenteric and subcutaneous adipose tissue, respectively. These liver, mesenteric and subcutaneous adipose tissue gene sets were subjected to enrichment analysis for retrieving their functional profiles (S1–S4 Tables). Due to the low number of differentially expressed genes in jejunum, we were unable to obtain a gene set enrichment signature for jejunum tissue. The top represented pathways in the liver included fatty acid elongation

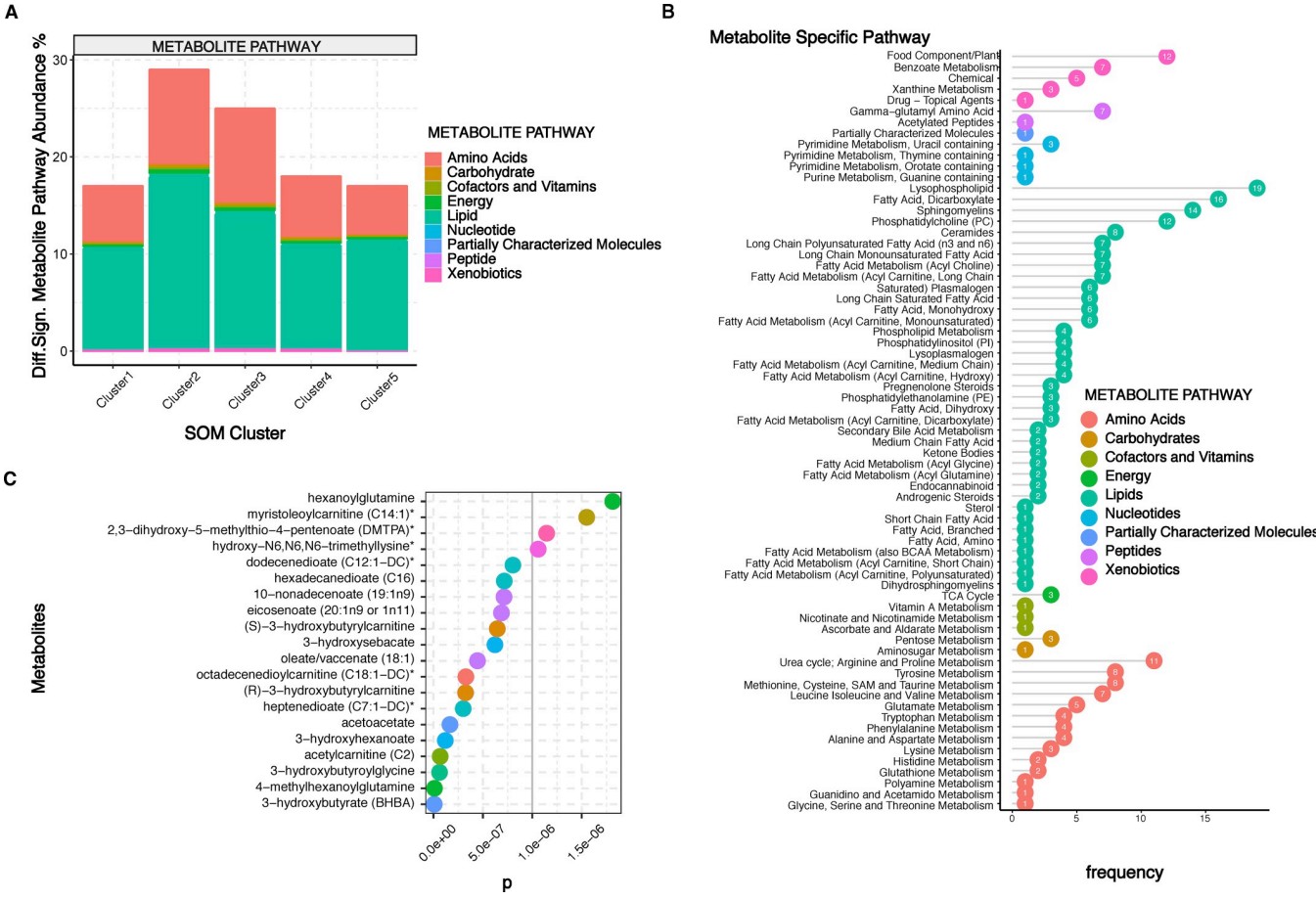

**Fig 2. Differentially abundant metabolites and metabolic pathways among the five defined SOM clusters (metabotypes). (A)** Relative abundance and distribution of differentially significant metabolites among SOM and k-means defined clusters. Clusters two and three are most abundant in lipids (especially lysophospholipids and sphingomyelins) and amino acids (urea, arginine and proline metabolism). **(B)** Distribution of differentially significant metabolic pathways among SOM and k-means defined clusters, where numbers within each dot indicate how many metabolites of that particular specific pathway were differentially abundant across clusters. **(C)** Top 20 differentially significant metabolites among the SOM and k-means defined clusters, (*P<0.05*).

/saturation reflecting lipids in the plasma, glycan and sphingolipid biosynthesis, cell function regulation (ErbB signaling pathway, protein export) and immune responses (Fig 3A).

The mesenteric adipose tissue was enriched for amino acid metabolic processes (Fig 3B) reflecting amino acids in the plasma, and subcutaneous adipose tissue was found enriched in many pathways related to pathogens (Fig 3C) and may reflect increased immune activation associated with metabolic disease. To investigate how these pathways are regulated across the five metabotypes, we examined the normalized gene expression levels of differentially expressed genes among the clusters. The metabolic pathways enriched within the hepatic transcriptome exhibited mixed directionality in regulation and were assessed individually, for each metabotype (Fig 3D). Amino acid metabolic pathways in mesenteric adipose tissue exhibited consistent upregulation in clusters 4 and 5 (Fig 3D). Transcriptome analysis from these three tissues showed distinct regulation of lipid, amino acid, immune response and pathogenic pathways amongst the metabotypes.

## Metabotypes exhibit distinct microbial community composition

Since the gut microbiota is known to be correlated with development of comorbidities linked to obesity [57–59], we also generated a gut microbiota profile for the BARIA individuals from

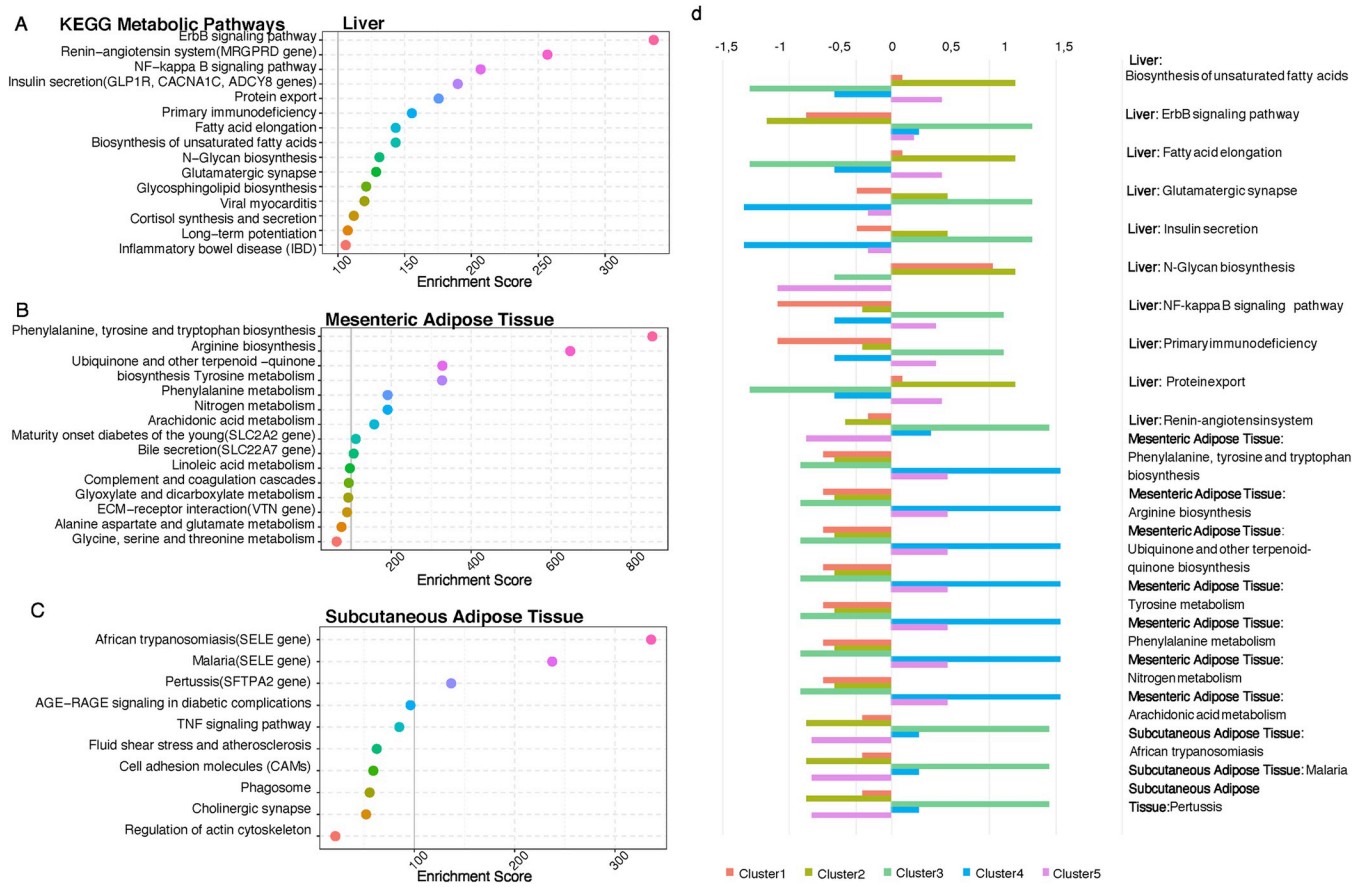

**Fig 3. Differentially enriched KEGG metabolic pathways among the five defined SOM clusters (metabotypes). (A)** Top 15 differentially enriched KEGG metabolic pathways for hepatic transcriptome among the SOM and k-means defined clusters, ranked based on their scores after differential gene expression analysis (DESeq2, *P<0.05*) and gene set analysis (GSA with EnrichR). **(B)** Top 15 differentially enriched KEGG metabolic pathways for mesenteric adipose transcriptome among the SOM and k-means defined clusters, ranked based on their scores after differential gene expression analysis (DESeq2, *P<0.05*) and gene set analysis (GSA) with EnrichR. **(C)** Top 10 differentially enriched KEGG metabolic pathways for subcutaneous adipose tissue transcriptome among the SOM and k-means defined clusters, ranked based on their scores after differential gene expression analysis (DESeq2, *P<0.05*) and gene set analysis (GSA with EnrichR). **(D)** 20 highest scoring KEGG metabolic pathways according to EnrichR GSA score for liver, mesenteric adipose and subcutaneous adipose tissues. *Z score* indicates different levels of differentially expressed pathways, for each SOM and k-means defined cluster.

shotgun metagenomic sequencing of fecal DNA. Statistical analysis revealed 119 differentially abundant species among the SOM metabotypes, the top 30 of which are shown in Fig 4A and are dominated by Bacteroidetes and Firmicutes, especially *Lactobacillus*.

Out of the 119 differentially abundant species, 70 belonged to Firmicutes phylum, 22 to Bacteroidetes, 11 to Actinobacteria, 11 to Proteobacteria, one to Chloroflexi, one to Cyanobacteria, one to Euryarchaeota, one to Spirochaetes and one to Fusobacteria. Within Firmicutes, *Clostridia* are more highly abundant for cluster 2 and *Weisella* for clusters 2, 4 and 5. Within Bacteroidetes, *Bacteroides* and *Prevotella* species are significantly more abundant in clusters 1, 2 and 3. For Actinobacteria, *Bifidobacterium* are considerably more abundant in 1 and 2, whereas species within Enterobacteriaceae family have higher abundance for clusters 4 and 5 (Fig 4B, S5 Table). In order to assess if there is a difference in alpha and beta diversity among metabotypes, we used a series of metrics (Observed, Chao1, ACE, Shannon, Simpson, Inverse Simpson for alpha diversity and Whittaker index along with dispersion analysis for beta diversity), shown in S3 and S4 Figs. Our metagenomics pipeline displayed that none of the different alpha diversity metrics reach statistical significance. The beta dispersion (with centroids)

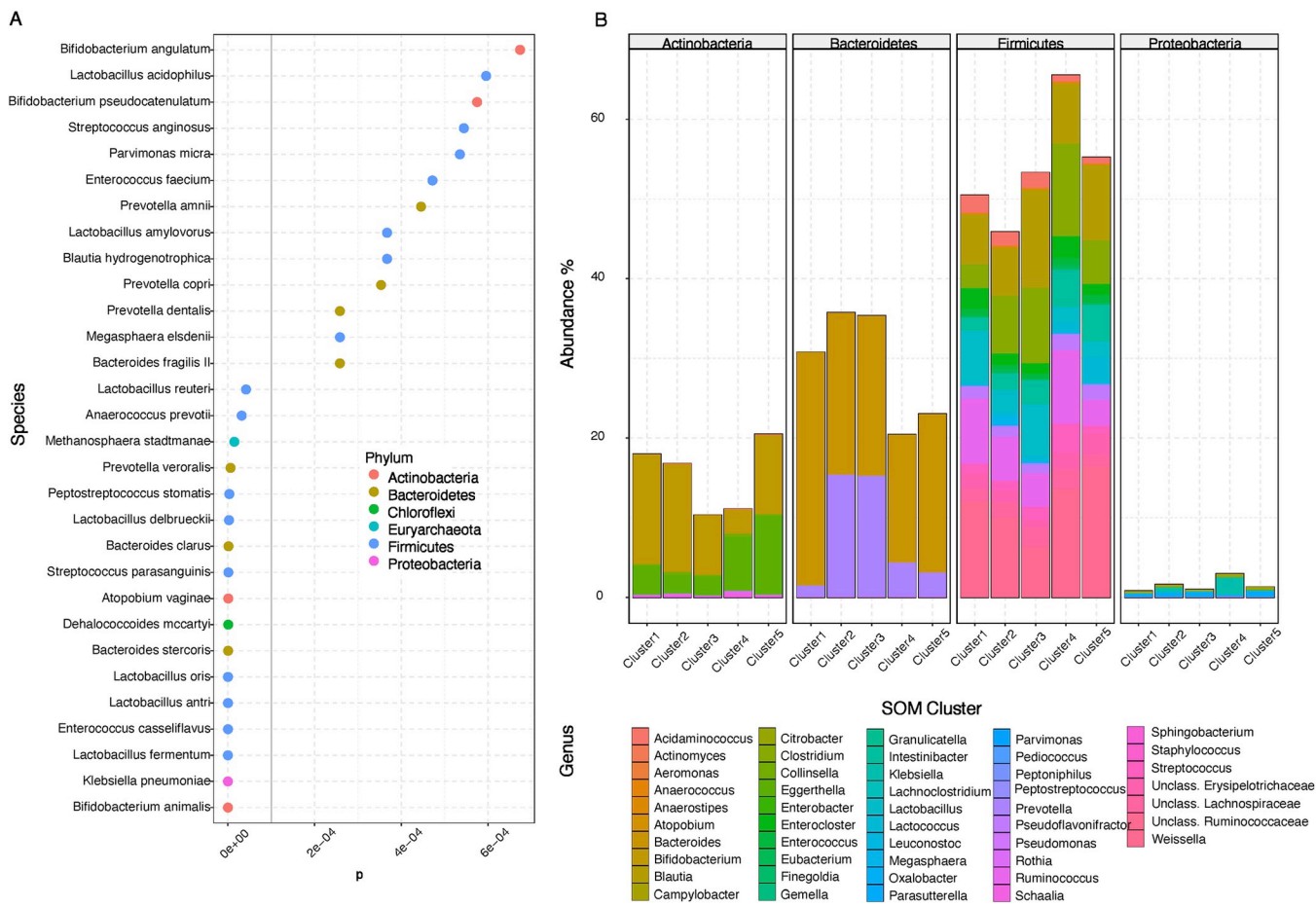

**Fig 4. Differentially significant microbial species and phyla among the five defined SOM clusters (metabotypes). (A)** Top 30 from 119 differentially significant microbial species among the SOM & k-means defined clusters, after differential analysis with DESeq2 (*P<0.05*). **(B)** Relative abundance and distribution of differentially significant microbial species for the top 4 most abundant phyla for SOM & k-means defined clusters.

results coupled with a permutational ANOVA (PERMANOVA) analysis (for 999 permutations) showed F = 0.19 and P = 0.9431. As seen in S4 Fig, the SOM-defined clusters largely spatially overlap but appear to have different centroids and different dispersions. Nevertheless, the large inter-individual variation cannot account for the negative PERMANOVA results, either. In such cases, there is a need to have a correct specification of the mean-variance relationship by means of multivariate extensions of GLM with methods such as negative binomials, DESeq2 [43]. The DESeq analysis revealed that despite the non-statistically significant diversities, there are SOM-defined clusters that are enriched in specific genera, such as *Bacteroides*, *Prevotella* and *Lactobacillus*.

As comparison, when the patients were grouped after presence or absence of metabolic syndrome, we only identified 54 differentially significant species. Similarly, none of the alpha diversity or beta diversity metrics or ordination were statistically significant. (S5 and S6 Figs). Out of the 54 significant gut microbial species 33 species belonged to Firmicutes phylum, 10 to Bacteroidetes, six to Actinobacteria, four to Proteobacteria and one to Fusobacteria (S7 Fig, S6 Table). Within Firmicutes, there is a trend for *Lactobacillus* species to be two to 8 times significantly less abundant in metabolic syndrome BARIA individuals. In contrast, statistically significant *Streptococcus* species are twice as abundant in metabolic syndrome diagnosed individuals. The majority of the gut microbial species belonging to Bacteroidetes is two to

three times depleted in metabolic syndrome diagnosed BARIA individuals, whereas Actinobacteria levels are elevated in metabolic syndrome. Differentially significant Proteobacteria tend to be depleted in metabolic syndrome.

Our analysis showed that metabolic syndrome diagnosis can indeed capture a fraction of the microbial variability within obesity. Even so, our suggested metabotyping approach can identify more gut microbial species across the spectrum of obesity and its related comorbidities.

## Individual metabotypes display unique clinical and multiple omics features

Our collective analyses show that the five different metabotypes clearly associate with unique gene expression and microbial community composition patterns and hence represents groups of individuals having distinct differences in their metabolism. To further explore these unique patterns, we next performed a detailed evaluation of the molecular fingerprints of each metabotype using the findings from the multiple omics datasets differential analysis.

17 individuals had metabotype 1 (13 women/four men), and they had the highest fat free mass 60.9 (54.1–93.8) kg and the highest total cholesterol (5.4 ± 1.1 mmol/L). Of these, four participants were treated for hypertension, three for T2D and four for GERD, whereas almost half the cluster's population (8 participants) was treated for high cholesterol. It is noticeable that three out of four male participants were co-treated for hypertension, GERD/*H. pylori* infection and cholesterol (Fig 1B). Isobutyrylcarnitine was at a higher level in this metabotype (see S6 Table) compared with the other metabotypes, and the same was observed for the tyrosine metabolic pathway intermediate 4-methoxyphenol. When associating the differentially significant fasting metabolites with anthropometric features (S8 and S9 Figs), we observed negative correlations between sphingomyelins, fasting glucose ($r = -0.8$, $P<0.001$), HbA1c ($r = -0.6$, $P<0.01$) and age ($r = -0.5$, $P<0.01$) specifically for this metabotype. In summary, metabotype 1 was characterized by high cholesterol, males using medication, downregulation of immune response pathways in the liver, lower abundance in *Prevotella* and higher abundance in *Bacteroides* (Fig 4B) compared to other metabotypes.

Metabotype 2 was the largest cluster consisting of 29 participants. It was female dominated (25 females/four males) and had the youngest individuals of 40 (20–57) years of average age with a BMI of 38.2 (32.9–60.9) kg/m$^2$. The highest number of T2D individuals was noted here (n = 8), with the highest mean HbA1c value at 42 ± 12 mmol/mol. The individuals were the most heavily medicated, since it contained 11 individuals with treatment for dyslipidemia, 6 with hypertension along with the individuals affected by T2D, of which four participants were treated for all conditions simultaneously. When considering the metabolome, lysophospholipids, 1-arachidonoyl-GPC*(20:4)*, 1-linoleoyl-GPC(18:3)*, 1-linoleoyl-GPE(18:2)*, 1-oleoyl-GPE(18:1), 1-palmitoyl-GPC(16:0)* and 1-stearoyl-GPC(16:0) were higher in comparison to the majority of the clusters. Similarly, branched-chain amino acid (BCAA) metabolites 1-carboxyethylvaline, 1-carboxyethylisoleucine and valine were all at elevated levels. 3-hydroxyoleoylcarnitine, 3-hydroxydecanoate and 2-hydroxybutyrate-2-hydroxyisobutyrate were positively correlated with both glucose and HbA1c ($r = 0.5$, $P<0.001$) (S8 and S9 Figs). To summarize, metabotype 2 represented the youngest individuals, yet the individuals being most heavily medicated for comorbidities. The individuals have high abundance of BCAAs and hydroxy fatty acids, even though fatty acids biosynthetic pathways were downregulated in mesenteric adipose fat. In the gut microbiome *Prevotella*, *Bacteroides* and *Lactobacillus* species were found to be highly abundant (Fig 4B).

There were 18 individuals having metabotype 3 and this metabotype had the highest percentage of males among the clusters (11 females/7 males). The individuals exhibited the

highest HOMA2-IR at 2.2 (0.5–4.7) and the highest HOMA2-β at 112 (52.7–226.2). It included three individuals with T2D (out of 6 in total treated for T2D), 10 hypertensive (out of 13 in total treated for hypertension) and 12 individuals treated for dyslipidemia (out of 15 in total treated for high cholesterol), whereas three were treated for T2D, hypertension and dyslipidemia at the same time. Even though the anthropometrics differed, the individuals of metabotype 3 had similar metabolome and microbiome profiles as metabotype 2, but with varying transcriptome patterns. Similar to metabotype 2, lysophospholipids, 1-arachidonoyl-GPC* (20:4)*, 1-linoleoyl-GPC (18:3)*, 1-linoleoyl-GPE (18:2)*, 1-oleoyl-GPE (18:1), 1-palmitoyl-GPC (16:0)* and 1-stearoyl-GPC (16:0) were detected in equally high levels in the individuals of metabotype 3. Noticeably, all sphingomyelins were elevated for individuals in this metabotype (S6 Table). Cluster 3 appeared to be the most insulin resistant and most treated for dyslipidemia, in spite of the highly abundant metabolome in lysophospholipids and sphingomyelins. In essence, hepatic upregulation of immune responses and subcutaneous adipose tissue upregulation of pathogenic-related pathways (Fig 3D), in conjunction with high *Prevotella* and *Lactobacillus* abundance (Fig 4B) completed the cluster's omics profile.

18 individuals had metabotype 4, including two individuals with T2D, 7 with hypertension and 8 treated for high cholesterol. The median age was the highest in this cluster compared to all others at 56 (39–64) years. Cluster 4 had the highest total body fat at 56.5 (40.6–78.9) kg. BARIA individuals stratified within this metabotype exhibited the highest creatinine at 75 (56–172) μmol/L, and lowest glomelular filtration rate at 78 (26–90) kl/1.73m$^2$. The transcriptomics datasets from liver tissues exhibited a very strong negative regulation of cortisol synthesis, glutamatergic synapse, cGMP-PKG signaling pathway and GABAergic synapse. When focusing on the gut microbial species, individuals of metabotype 4 had many changes in the microbial composition, and the abundance of some of the species correlated with plasma glucose, low-density lipoprotein (LDL) cholesterol and cholesterol (S10 Fig). In outline, the individuals of metabotype 4 had potentially impaired kidney function, high body fat, downregulation of synaptic pathways in the liver, upregulation of fatty acid metabolic process in mesenteric adipose tissue, upregulation of pathogenic-related pathways in subcutaneous adipose tissue and increased levels of *Clostridium*, *Streptococcus* and *Klebsiella* in the gut microbial metagenome.

17 individuals had metabotype 5, with dominance of females (14 females/three males) and the median age of the individuals in this cluster was the second youngest, 44 (22–62) years. The participants were relatively treatment naïve, only four were treated for T2D, three for hypertension and three participants for dyslipidemia, with very little overlapping treatments. 1-carboxyethylvaline, 1-carboxyethylisoleucine and valine positively correlated with HbA1c ($r = 0.4$, $P<0.01$) (S8 and S9 Figs). In conclusion, metabotype 5 corresponded to a relatively young cluster, with no striking comorbidity treatment, upregulated fatty acid metabolic pathways and immune response pathways in the liver and highly abundant in *Citrobacter*.

In order to ensure that this analysis of the multiple omics datasets is not confounded by covariates such as age and gender, we adjusted our statistical analysis to account for these two factors, in addition to our previous analysis. For the metabolome, we observed 174 metabolites being differentially significant among SOM-defined clusters, instead of 226 that we had before. In spite of that, the relative distribution of differentially significant pathways and their % pathway abundance remains identical (S11 Fig). As far as the Metabolite Specific Pathways is concerned, the same pathways appear to be most abundant, such as lysophospholipids, phosphatidylcholines, dicarboxylate fatty acids, and branched-chain amino acid metabolites and also maintain their distribution among the metabotypes.

In the hepatic and adipose tissue transcriptome analysis, we noticed a direct effect from the confounders. This is expected since even if the genetic making(genome) of men and women is the same, the transcriptome is distinctly dimorphic with dissimilar disease susceptibilities [60].

Pathogenic pathways along with compound degradation pathways were more dominant in the liver transcriptome enrichment, especially in clusters 1 and 2 (S12 Fig). In general, after the confounders removal there is a more distinct pattern of nutrients catabolism, degradation and absorption and less presence of inflammatory pathways in the hepatic transcriptome when comparing to the pathways that we obtained in our previous enrichment analysis. Surprisingly, the mesenteric adipose tissue was once more enriched for amino acid metabolic processes, in fact the same pathways as before including phenylalanine, tyrosine, tryptophan biosynthesis and fatty acids metabolism. In subcutaneous adipose tissue we also obtained a different set of enriched metabolic pathways, this time not including pathogenic pathways. Instead, we observed similar pathways as in the mesenteric adipose tissue, primarily amino acid and fatty acid metabolic pathways.

In the metagenomics dataset analysis 109 gut microbial species being differentially significant among SOM-defined clusters, instead of 288 that we had before. However, removing those confounders does not affect the overall profiling of the dataset, especially in the most dominant Phyla of Bacteroidetes and Firmicutes (S13 Fig). Even after the model adjustment, *Prevotella*, *Bacteroides and Lactobacillus* remained the most dominant species with exactly the same distribution among the SOM-defined metabotypes.

## Multi-omics integration elucidates discriminatory signature and associations between datatypes

To reveal key interactions between multi-omics data sets, we used DIABLO [53] to identify how the five metabotypes are associated with altered expression in different tissues and an altered gut microbiota. Initially, we provided the differentially abundant metabolites, genes from liver, jejunum, mesenteric and subcutaneous adipose tissue, and gut microbial species for each BARIA individual, along with their respective metabotype membership, as input to the algorithm. DIABLO simultaneously calculates the correlations among all input multiple omics datasets and selects a minimal set of input variables that differentiate the metabotypes. The computational framework used here for integrating various omics datasets successfully identified a highly correlated discriminatory signature for SOM-defined obesity phenotypes that includes multiple *Prevotella* species (*P. veroralis*, *P. copri*, *P. multisaccharivorax*, *P. oulorum*, *P. denticola*, *P.* sp. oral taxon 299, *P. bryantii*, *P. melaninogenica*), *Intestinibacter bartlettii*, *Anaerococcus prevotii*, lipid metabolites (especially phospatidylcholines), *hepatic function associated genes*, lipid metabolism and cardiomyopathy pathways, subcutaneous adipose tissue *IL6* and *SELE* genes involved in inflammatory and immune system pathways and mesenteric adipose tissue genes enriched in prolactin signaling, T2D and PI3K-Akt signaling pathways (S14 Fig).

## Metabotypes are associated with weight loss response to bariatric surgery

In order to define the clinical value of metabotyping, we had to assess the metabotypes' response to bariatric surgery. Hence, we performed a longitudinal biometrics post-operative control of the BARIA obese individuals at three time points: three months, six months and 12 months after surgery, where we monitored the weight, waist circumference and upper leg circumference. It is noteworthy that there are no distinct statistically significant responses in the weight loss or waist circumference reduction immediately after bariatric surgery (3 months after surgery), contrary to what would be expected (Fig 5A and 5B).

There is a trend that metabotypes 2 and 5 have the highest weight loss one year post-operatively (35kg and 38 kg in average, respectively). Metabotype 2 exhibits the largest waist circumference loss at three months after surgery (12cm) even if this is not deemed statistically significant (Fig 5C and 5D). However, there is a clear pattern in the reduction of adipose tissue

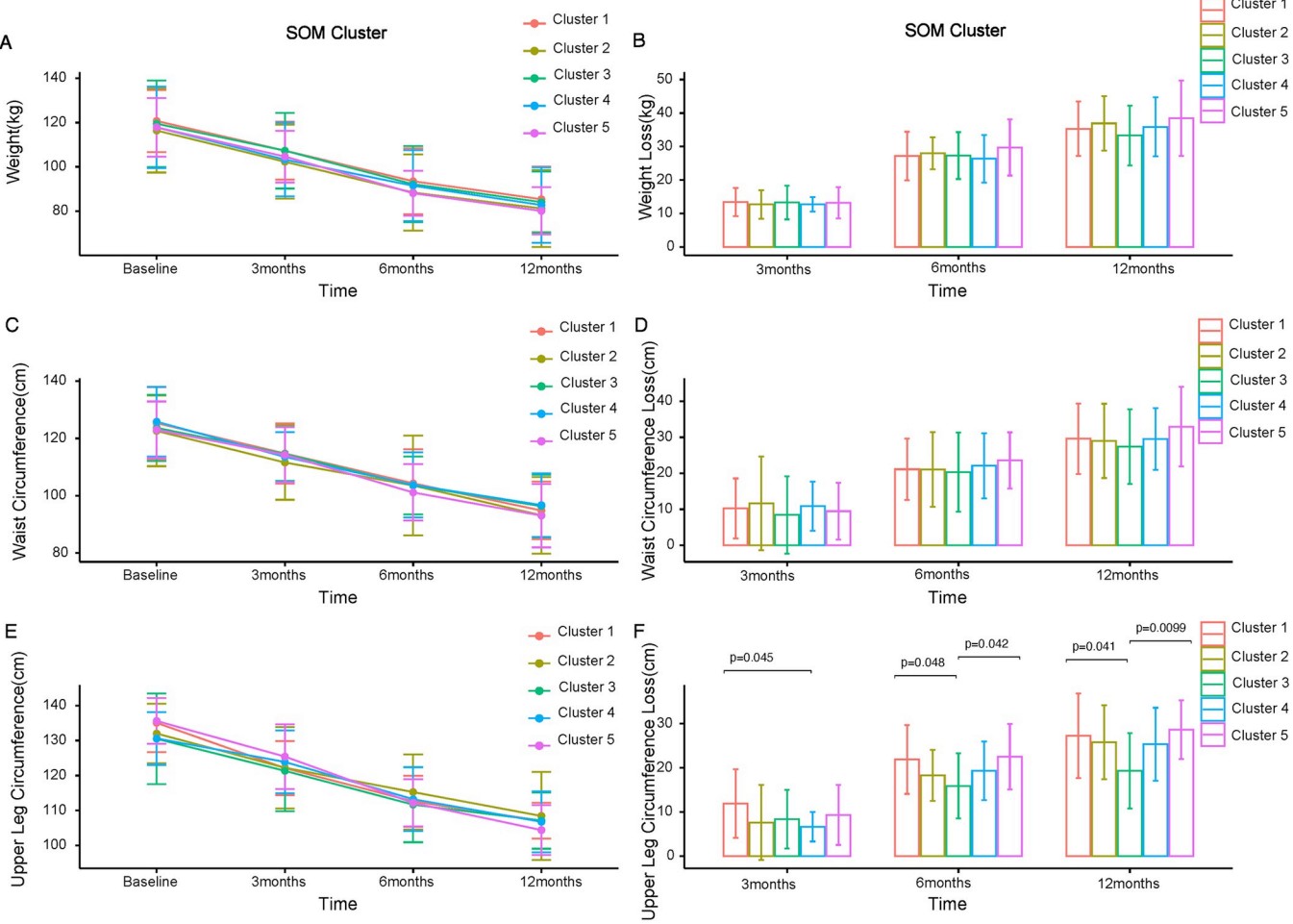

**Fig 5. Weight and fat loss progression at distinct time points after bariatric surgery for the five defined SOM clusters (metabotypes). (A)** Weight (kg) of BARIA individuals at baseline, three months, six months and one year after bariatric surgery for each metabotype. **(B)** Weight loss(kg) of BARIA individuals at baseline, three months, six months and one year after bariatric surgery for each metabotype. **(C)** Waist circumference (cm) of BARIA individuals at baseline, three months, six months and one year after bariatric surgery for each metabotype. **(D)** Reduction of waist circumference(cm) of BARIA individuals at baseline, three months, six months and one year after bariatric surgery for each metabotype. **(E)** Upper leg circumference (cm) of BARIA individuals at baseline, three months, six months and one year after bariatric surgery for each metabotype. **(F)** Reduction of upper leg circumference(cm)of BARIA individuals at baseline, three months, six months and one year after bariatric surgery for each metabotype. Statistical significance among metabotypes is calculated with t-test and adjusted with FDR; the symbols indicating significance among metabotypes are '*': *P< = 0.05*, '**': *P< = 0.01*, '***': *P< = 0.001*.

in the upper leg circumference. Metabotypes 1 and 5 are the best responders when it comes to upper leg circumference reduction, with the loss being consistent at all three time points. Upper leg circumference loss is significant (*P<0.05*) when compared to the worst responder cluster, metabotype 3 (Fig 5E and 5F). This trend is the same for weight loss, regardless of being confirmed by statistical calculations.

Surprisingly, when the BARIA individuals were grouped according to having or not-having metabolic syndrome, there were no notable statistically significant differences in weight and adiposity loss in none of the three time points.

## Discussion

Here we present a novel unsupervised machine learning framework for stratification of individuals in human volunteer cohorts, with a high prevalence and variance of comorbidities.

This framework enables a naïve to clinical labels stratification based on fasting metabolome rather than purely clinical parameters that may fail to accurately encompass the multitude of nuances in human population-based studies. The main findings of our study revealed pronounced changes in lysophospholipids, phosphatidylcholines, dicarboxylate fatty acids, sphingomyelins, and branched-chain amino acid metabolites among the five different metabotypes; KEGG metabolic pathways related to immune functions, fatty acid biosynthesis and elongation, protein signaling and pathogenic pathways were regulated in different ways for each metabotype; the abundance of *Prevotella* and *Lactobacillus* species varied the most between the metabotypes, and metabotypes 4 and 5 had a lower abundance compared to metabotypes 2 and 3. Multi-omics integration enabled reducing the dimensionality and identified a concrete biomarker signature able to differentiate between the five distinct metabotypes. The differences in metabolism among the individuals in the five metabotypes are associated with different responses in terms of weight loss and reduction of waist and upper leg circumference to bariatric surgery.

A considerable advantage of our approach is that SOM and k-means clustering effectively reduced the initial omics dimensionality and resulted in a reusable topological projection of the metabolome. Given the lack of an external multimodal multiple omics dataset for validating our results, establishing a metabolome mapping that can recognize or characterize new unknown inputs can be proven useful. New metabolomes can be projected into the same map, without the requirement of further algorithmic training. That way we can compare metabolic distances among new BARIA inclusions or even the potential post-surgical metabolomes of the initial 106 inclusions. Comparing the post-operative metabolome with the baseline pre-operative state could provide further mechanistic comprehension of the pathophysiological mechanisms of obesity and the responses to the bariatric surgery intervention in the future. Also using the multi-parameter metabolic syndrome as a classifier was here shown not to enable new insight into what drives differences in metabolism within the cohort. Metabotyping identified more gut microbial species among BARIA individuals, whereas the metabolic syndrome classification captured a fraction of the microbial variability. It has been previously attempted in animal studies to model interactions between genes, gut microbiota and the molecular mechanisms underlying obesity [61–63], but their clinical application to humans has been limited [64] so far. Increased microbial variability among metabotypes along with the results from the KEGG pathways enrichment in liver and adipose tissues could be the effect of gut microbial species in the hosts' gene regulation. In the metabotype comparison, the statistically significant anthropometric features of age and glomerular filtration rate along with the differentially significant KEGG pathways could plausibly reflect the process of cellular and biological senescence [65,66]. The detection of senescence in the metabolome by our proposed SOM and k-means methodology, without prior knowledge of biometric characteristics strengthens our claim that the identified metabotypes stand as different representations of human metabolism among the BARIA obese individuals.

Our findings reveal that the overall effect of the metabotyping is still present in both the metabolome and the gut microbial metagenome even after regressing out confounders like age and gender. What has changed after the correction, is that metabolite compounds and gut microbial species that were less abundant in the initial dataset have now been eliminated after confounder adjustment.

On the other hand, confounding factors appear to have a more direct effect on the transcriptome, even if enrichment analysis exhibited the distinct regulation of lipid, amino acid, and pathogenic pathways amongst the metabotypes.

A limiting factor that needs to be considered when interpreting our findings is the selection of the eligible individuals for bariatric surgery. The significant variability within human

cohorts is often not possible to capture in a finite number of clinical variables. For example, classifiers for obesity-associated comorbidities such as hypertension, T2D, and dyslipidemia may be treated as binary variables (present *vs*. absent) [9], however the overall wellness of an individual with any of these disorders can vary significantly as a function of how well managed each of these conditions are, among many other factors. The BARIA exclusion criteria for surgical interventions have to be strictly met for minimizing the risks and complications of such an evasive procedure. As a result, the BARIA inclusions might not fully represent the obesity spectrum. Many of differentially significant metabolites that were identified by our pipelines are directly implicated in inflammatory pathways. A clinical inflammatory marker, such as C-Reactive Protein (CRP) would be very valuable for confirming this observation. However, CRP at baseline did not exhibit any overall statistical significance among the SOM-defined clusters and was not available in the one year follow up (S15 Fig). There is a visible trend in weight loss and leg/waist circumference reduction among the SOM and k-means defined clusters over time. Nonetheless the statistically significant differences among all the identified SOM clusters were not conclusive, probably due to statistical power. Despite the 106 BARIA inclusions and the high quality of the omics dataset, each cluster contains 17–29 individuals, which might account for the values of the statistically significant results.

## Conclusions

The principal contribution of this study is the detailed omics dataset for obese individuals, that includes metabolome, microbiome and especially transcriptome from multiple tissues. Our findings suggested that participants' stratification based on metabotyping could enhance our ability to get molecular insights into the causes of diseases from multi-omics integrative analysis. The combination of SOM metabotyping and DIABLO correlation analysis highlights the data-driven nature of this approach. DIABLO analysis enabled the identification of an underlying common yet discriminatory minimal multi-omics signature for the SOM-defined metabotypes, that could lead to predictive markers of the bariatric surgery outcome. In this light, use of biologic parameters such as the plasma metabolome, as a direct readout of the overall status of an entire multiorgan system host and its microbiome, to determine grouping of individuals, offers a unique approach that may more accurately classify individuals into distinct disease physiological states [11,12,67]. Rather than traditional clinical disease classifiers, this grouping method may reduce the confounding effects of such clinical metadata [68,69]. The multiple omics dataset's association framework can be the starting point for selecting candidate compounds for a more thorough examination and provide mechanistic insight into the causality of pathogenicity originating in the tissues, mediated by bacteria and materializing via metabolites and clinical metadata. The multi-omics integrative framework implemented could also be utilized as a hypothesis generating tool for comprehending cardiometabolic disease. Our data suggest that self-organized metabotyping, based only on metabolite distribution, with no other prior knowledge on the individuals' clinical status in combination with DIABLO integrative analysis, constitute a valuable computational approach studying multifaceted metabolic disorders.

## Supporting information

**S1 Fig. Clinical variables distribution between BARIA individuals diagnosed with and without Metabolic Syndrome. (i)** Fasting glucose **(ii)** Hba1c, **(iii)** HOMA2_IR, **(iv)** Systolic Blood Pressure, **(v)** Triglycerides and **(vi)** HDL cholesterol statistical significance calculated with Kruskal-Wallis test, symbols indicating significance among metabotypes: '*': *P< = 0.05*, '**': *P< = 0.01*, '***': *P< = 0.001*.
(TIF)

**S2 Fig. Differentially significant metabolites between BARIA individuals diagnosed with and without Metabolic Syndrome.** Statistical analysis conducted with HybridMTest and adjusted p value with Estimated Bayesian Probability (*P<0.05*).
(TIF)

**S3 Fig. Different measures of microbial species alpha diversity (Observed, Chao1, ACE, Shannon, Simpson, Inverse Simpson) for gut microbial species species among the SOM clusters (metabotypes).** None of the alpha diversity metrics reach statistical significance.
(TIF)

**S4 Fig. Beta diversity (Whittaker measure) and homogeneity of multivariate dispersions for gut microbial species species among the SOM clusters (metabotypes).** Average Euclidean distances in principal coordinate space between the samples and their respective group centroid are: Cluster1 = 0.3946, Cluster2 = 0.3828, Cluster3 = 0.3777, Cluster4 = 0.3718, and Cluster 5 = 0.3769. The Whittaker diversity index did not reach statistical significance.
(TIF)

**S5 Fig. Differentially significant microbial species and alpha diversity between BARIA individuals diagnosed with and without Metabolic Syndrome. (a)** Different measures of microbial species alpha diversity (Observed, Chao1, ACE, Shannon, Simpson, Inverse Simpson) and none of the alpha diversity metrics reach statistical significance. **(b)** Differentially significant microbial species between BARIA individuals diagnosed with and without Metabolic Syndrome, after statistical analysis with DESeq2 (*P<0.05*).
(TIFF)

**S6 Fig. Beta diversity (Whittaker measure) and homogeneity of multivariate dispersions for gut microbial species species between BARIA individuals diagnosed with and without Metabolic Syndrome.** Average Euclidean distances in principal coordinate space between the samples and their respective group centroid are: MetabolicSyndrome = 0.3776 and NoMetabolicSyndrome = 0.3992. The Whittaker diversity index did not reach statistical significance.
(TIF)

**S7 Fig. Relative abundance and distribution of differentially significant microbial species between BARIA individuals diagnosed with and without Metabolic Syndrome. (a)** Relative abundance and phyla composition of the differentially significant microbial between BARIA individuals diagnosed with and without Metabolic Syndrome. **(b)** Distribution of differentially significant microbial species across phyla between BARIA individuals diagnosed with and without Metabolic Syndrome.
(TIF)

**S8 Fig. Spearman correlation between the first 145 differentially significant metabolites and clinical variables among the SOM clusters (metabotypes).** Significance codes are '***' 0.001 '**' 0.01 '*' 0.05 '.' 0.1 ' ' 1.
(TIFF)

**S9 Fig. Spearman correlation between the last 144 differentially significant metabolites and clinical variables among the SOM clusters (metabotypes).** Significance codes are '***' 0.001 '**' 0.01 '*' 0.05 '.' 0.1 ' ' 1.
(TIFF)

**S10 Fig. Spearman correlation between differentially significant gut microbial species and clinical variables among the SOM clusters (metabotypes).** Significance codes are '***' 0.001

'**' 0.01 '*' 0.05 '.' 0.1 ' ' 1.
(TIFF)

**S11 Fig. Differentially abundant metabolites and metabolic pathways among the five defined SOM clusters (metabotypes) adjusted for the confounding factors of age and gender. (A)** Relative abundance and distribution of differentially significant metabolites among SOM and k-means defined clusters. Clusters two and three are most abundant in lipids (especially lysophospholipids and sphingomyelins) and amino acids (urea, arginine and proline metabolism). **(B)** Distribution of differentially significant metabolic pathways among SOM and k-means defined clusters, where numbers within each dot indicate how many metabolites of that particular specific pathway were differentially abundant across clusters. **(C)** Top 20 differentially significant metabolites among the SOM and k-means defined clusters, (*P<0.05*).
(TIFF)

**S12 Fig. Differentially enriched KEGG metabolic pathways among the five defined SOM clusters (metabotypes)), adjusted for the confounding factors of age and gender. (A)** Top 15 differentially enriched KEGG metabolic pathways for hepatic transcriptome among the SOM and k-means defined clusters, ranked based on their scores after differential gene expression analysis (DESeq2, *P<0.05*) and gene set analysis (GSA with EnrichR). **(B)** Top 15 differentially enriched KEGG metabolic pathways for mesenteric adipose transcriptome among the SOM and k-means defined clusters, ranked based on their scores after differential gene expression analysis (DESeq2, *P<0.05*) and gene set analysis (GSA) with EnrichR). **(C)** Top 10 differentially enriched KEGG metabolic pathways for subcutaneous adipose tissue transcriptome among the SOM and k-means defined clusters, ranked based on their scores after differential gene expression analysis (DESeq2, *P<0.05*) and gene set analysis (GSA with EnrichR). **(D)** 20 highest scoring KEGG metabolic pathways according to EnrichR GSA score for liver, mesenteric adipose and subcutaneous adipose tissues. *Z score* indicates different levels of differentially expressed pathways, for each SOM and k-means defined cluster.
(TIFF)

**S13 Fig. Differentially abundant metabolites and metabolic pathways among the five defined SOM clusters (metabotypes), adjusted for the confounding factors of age and gender. (A)** Relative abundance and distribution of differentially significant metabolites among SOM and k-means defined clusters. Clusters two and three are most abundant in lipids (especially lysophospholipids) and amino acids (urea, arginine and proline metabolism). **(B)** Distribution of differentially significant metabolic pathways among SOM and k-means defined clusters, where numbers within each dot indicate how many metabolites of that particular specific pathway were differentially abundant across clusters. **(C)** Top 20 differentially significant metabolites among the SOM and k-means defined clusters, (*P<0.05*).
(TIFF)

**S14 Fig. DIABLO analysis and correlations among multiple omics datasets for the five defined SOM clusters. (a)** Total correlation matrix for the differentially significant metabolites, genes, microbial species from all the different omic datasets after Sparce Principal Least Squares Regression with mixOmics DIABLO. Highest total dataset correlation was observed for differentially significant genes from the liver RNASeq dataset and mesenteric adipose tissue at r = 0.89, followed by the metabolomics dataset with the liver RNASeq at r = 0.57 and the metagenomics dataset with both liver and mesenteric adipose tissue RNASeq at r = 0.53. **(b)** Circular correlation plot by DIABLO, for selecting top contributing components from each omics dataset (metabolites, genes, bacterial species). Correlation cut-off was r = 0.7. The chosen elements constituted a highly correlated discriminatory signature for the five metabotypes.

This signature involves a series of: i) *Prevotella* species (*P. veroralis*, *P. copri*, *P. multisacchari-vorax*, *P. oulorum*, *P. denticola*, *P. sp. oral taxon 299*, *P.bryantii*, *P. melaninogenica*), *Intestinibacter bartlettii*, *Anaerococcus prevotii;* ii) lipid metabolites (especially phospatidylcholines); iii) liver genes enriched in oxidative phosphorylation, lipid metabolism and cardiomyopathy pathways; iv) subcutaneous adipose fat *IL6* and *SELE* genes involved in inflammatory and immune system pathways; v) mesenteric adipose fat genes enriched in prolactine signaling, T2DM and PI3K-Akt signaling pathways.
(TIFF)

**S15 Fig. C-Reactive protein associated with obesity and its statistical significance across the metabotypes (SOM & k-means defined clusters.** Statistical significance among metabotypes is calculated with Kruskal-Wallis test and p value has been adjusted with FDR.
(TIFF)

**S1 Table.  A.** Differentially significant genes in liver tissue among the 5 SOM clusters (metabotypes). ENSGeneID: Ensemble Gene ID Identifier, MeanCluster: Mean gene expression in a particular cluster, padj: Log Ratio Test adjusted p value, as calculated by DESeq2. **B.** Differentially significant genes in mesenteric adipose tissue among the 5 SOM clusters (metabotypes). ENSGeneID: Ensemble Gene ID Identifier, MeanCluster: Mean gene expression in a particular cluster, padj: Log Ratio Test adjusted p value, as calculated by DESeq2. **C. Differentially significant genes in subcutaneous adipose tissue among the 5 SOM clusters (metabotypes).** ENSGeneID: Ensemble Gene ID Identifier, MeanCluster: Mean gene expression in a particular cluster, padj: Log Ratio Test adjusted p value, as calculated by DESeq2.
(XLSX)

**S2 Table. Differentially significant gene set enrichment results for all tissues.** DEGGroup: Groups assigned by DEG Report software, representing genes with similar expression levels enriched in KEGG pathways, Cluster_DEGScore: z score assigned by DEG Report software, representing the expression levels of enriched KEGG pathways for each cluster, p value: Wilcoxon p value assigned by Enrichr software indicating the statistical significance of the enriched KEGG pathway, EnrichrCombinedScore: Score assigned by Enrichr software indicating the significance of the enriched KEGG pathway.
(XLSX)

**S3 Table. Differentially significant gut microbial species among the 5 SOM clusters (metabotypes).** MeanMicrobialSpeciesAbundanceCluster: Mean microbial species abundance in a particular cluster, padj: Log Ratio Test adjusted p value, as calculated by DESeq2.
(XLSX)

**S4 Table. Differentially significant metabolites among the 5 SOM clusters (metabotypes).** MeanMetaboliteAbundanceCluster: Mean metabolite abundance in a particular cluster, padj: Estimated Bayesian Probability adjusted p value.
(XLSX)

**S5 Table. DIABLO all multi-omics datasets correlation matrix.**
(XLSX)

**S6 Table. Differentially significant gut microbial species between BARIA individuals diagnosed with Metabolic Syndrome and with No Metabolic Syndrome.** padj: Log Ratio Test adjusted p value, as calculated by DESeq2.
(XLSX)

## Acknowledgments

We thank SNIC and C3SE for their technical assistance in our throughout the duration of BARIA project.

## Author Contributions

**Conceptualization:** Fredrik Bäckhed, Jens Nielsen.

**Data curation:** Dimitra Lappa, Lisa M. Olsson.

**Formal analysis:** Dimitra Lappa.

**Investigation:** Dimitra Lappa.

**Methodology:** Dimitra Lappa.

**Project administration:** Louise E. Olofsson.

**Resources:** Dimitra Lappa, Abraham S. Meijnikman, Kimberly A. Krautkramer, Lisa M. Olsson, Ömrüm Aydin, Anne-Sophie Van Rijswijk, Yair I. Z. Acherman, Maurits L. De Brauw, Valentina Tremaroli, Louise E. Olofsson, Annika Lundqvist, Victor E. A. Gerdes.

**Software:** Lisa M. Olsson.

**Supervision:** Thue W. Schwartz, Max Nieuwdorp, Fredrik Bäckhed, Jens Nielsen.

**Writing – original draft:** Dimitra Lappa.

**Writing – review & editing:** Abraham S. Meijnikman, Kimberly A. Krautkramer, Ömrüm Aydin, Anne-Sophie Van Rijswijk, Yair I. Z. Acherman, Maurits L. De Brauw, Valentina Tremaroli, Louise E. Olofsson, Annika Lundqvist, Siv A. Hjorth, Boyang Ji, Victor E. A. Gerdes, Albert K. Groen, Thue W. Schwartz, Fredrik Bäckhed, Jens Nielsen.

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
