## [Decision Letter · Decision Letter 0]

30 Sep 2022

PONE-D-22-23260Self-organized metabotyping of obese individuals identifies clusters responding differently to bariatric surgeryPLOS ONE

Dear Dr. Nielsen,

Thank you for submitting your manuscript to PLOS ONE. After careful consideration, we feel that it has merit but does not fully meet PLOS ONE’s publication criteria as it currently stands. Therefore, we invite you to submit a revised version of the manuscript that addresses the points raised during the review process.

We look forward to receiving your revised manuscript.

Kind regards,

Priyadarshini Kachroo

Academic Editor

PLOS ONE

Journal Requirements:

"The BARIA study is funded by the Novo Nordisk Foundation (NNF15OC0016798). The Novo Nordisk Foundation Center for Basic Metabolic Research is supported by an unconditional grant (NNF10CC1016515) from the Novo Nordisk Foundation to University of Copenhagen. The BARIA study is a Scandinavian-Dutch collaboration. Funding from Knut and Alice Wallenberg Foundation is also acknowledged."

"The computations and RNA Sequencing were enabled by resources provided by the Swedish National Infrastructure for Computing (SNIC) at C3SE (SNIC Computational Center of Chalmers University of Technology) partially funded by the Swedish Research Council through grant agreement no. 2018-05973."

"The BARIA study is funded by the Novo Nordisk Foundation (NNF15OC0016798). The Novo Nordisk Foundation Center for Basic Metabolic Research is supported by an unconditional grant (NNF10CC1016515) from the Novo Nordisk Foundation to University of Copenhagen. The BARIA study is a Scandinavian-Dutch collaboration. Funding from Knut and Alice Wallenberg Foundation is also acknowledged."

"The BARIA study is funded by the Novo Nordisk Foundation (NNF15OC0016798). The Novo Nordisk Foundation Center for Basic Metabolic Research is supported by an unconditional grant (NNF10CC1016515) from the Novo Nordisk Foundation to University of Copenhagen. The BARIA study is a Scandinavian-Dutch collaboration. Funding from Knut and Alice Wallenberg Foundation is also acknowledged."

Please state what role the funders took in the study.  If the funders had no role, please state: ""The funders had no role in study design, data collection and analysis, decision to publish, or preparation of the manuscript."" If this statement is not correct you must amend it as needed. 

7. Your ethics statement should only appear in the Methods section of your manuscript. If your ethics statement is written in any section besides the Methods, please move it to the Methods section and delete it from any other section. Please ensure that your ethics statement is included in your manuscript, as the ethics statement entered into the online submission form will not be published alongside your manuscript. 

8. Please upload a new copy of Figure 2 and Supporting Figures 5, 8, 9, 10 and 11 as the detail is not clear. Please follow the link for more information: 

https://blogs.plos.org/plos/2019/06/looking-good-tips-for-creating-your-plos-figures-graphics/

https://blogs.plos.org/plos/2019/06/looking-good-tips-for-creating-your-plos-figures-graphics/

Reviewers' comments:

Reviewer's Responses to Questions

**Comments to the Author**

1. Is the manuscript technically sound, and do the data support the conclusions?

Reviewer #1: Partly

Reviewer #2: Yes

2. Has the statistical analysis been performed appropriately and rigorously? 

Reviewer #1: Yes

Reviewer #2: Yes

3. Have the authors made all data underlying the findings in their manuscript fully available?

Reviewer #1: Yes

Reviewer #2: No

4. Is the manuscript presented in an intelligible fashion and written in standard English?

Reviewer #1: Yes

Reviewer #2: Yes

5. Review Comments to the Author

Reviewer #1: This study performs a distinctive analysis in understanding the response in obesity to bariatric surgery, integrating gut microbiomics and plasma metabolome via the usage of SOMs, and further informing these profiles with tissue transcriptomics. Whilst this multi-omics strategy is conducted on a relatively small cohort of 106, the statistics are rigorous and provide a uniquely informed disease stratification.

1) Table 1 – please include a column listing demographics without being separated into the SOMs. This will help to understand the distribution of the cohort amongst the SOMs.

2) Line 148- 151: “652 metabolites were fully detected across all samples, 640 metabolites were partially detected across all samples, and 53 metabolites were not detected or failed to reach detection limit and therefore had a missing value”. Please clarify on the limit of detection here; the phrasing can be misinterpreted. What is the mean CV for those metabolites partially detected? Additionally, please clarify if these are relative or absolute abundances.

3) Line 483: “dominated by Bacteroidetes and Firmicutes” – these genera are often increased in abundance in ageing. Given the acute age distribution of the cohort, have the authors considered this as confounding in this study of obesity? Line 566, states that “metabotype 2 represented the youngest individuals, yet the individuals being most heavily medicated for comorbidities…. In the gut microbiome Prevotella, Bacteroides and Lactobacillus species were found to be highly abundant”. Could these metagenome associated metabotypes be driven solely by age and sex? Given the small sample size, this is likely to have an impact on modelling. Please adjust models to correct for age and sex, and run training models to investigate the impact of age and sex on observed phenotypes.

4) Line 628- “liver genes enriched in oxidative phosphorylation”. Please correct to ‘hepatic function associated genes’.

5) Line 640- “It is noteworthy that there are no distinct statistically significant responses in the weight loss or waist circumference reduction immediately after bariatric surgery (3 months after surgery), contrary to what would be expected”. In the methods the authors have listed clinical lab biochemistry measures, focusing more so on glucose and triglycerides. This would be important in understanding the clinical drivers of the metabotypes. It is stated, “pronounced changes in lysophospholipids, phosphatidylcholines, dicarboxylate fatty acids, sphingomyelins, and branched-chain amino acid metabolites among the five different metabotypes” – all of these metabolites are all implicated in inflammation. Have the authors looked at inflammatory markers such as CRP, and if that is lacking looking at red cell distribution width (RDW) as a proxy for inflammation?

Reviewer #2: This is a well-written manuscript using omics data to predict long-term weight loss. The authors used plasma metabolomics, adipose tissue gene expression data, and fecal metagenomics from individuals underwent bariatric surgery. The authors identified distinct metabotypes associated with immune function, fatty acid metabolism, protein signaling and obesity pathogenesis. The gut metagenome data identified metabotype associated with cardiometabolic comorbidities. However, several points need to be addressed by authors before considering for publication.

1. The manuscript is not clear whether you are using multi-omics or multiple types of omics data. The methods and results sections need to be re-written to clearly discuss this concept and the results associated.

2. Figure 1b, some clusters were associated with gender, and other characters not related to weight loss. Please regress out by taking the residuals confounders from the omics matrix.

3. Figure 5a, 5b, 5c, 5d, 5e, it seems the clusters didn't separate out well long-term weight loss, please provide statistics.

4. This data leveraged many features, and no discussion of multiple testing / ways to control for multiple testing. If it is based on nominal significance, please state in the abstract, results, and discussion.

5. Again, many of the results discussed clusters that reflect gender, age and etc. Please take out variations of other confounders by regressing out these variables.

6. The resolution of the figures are not good, please modify.

6. PLOS authors have the option to publish the peer review history of their article (what does this mean?). If published, this will include your full peer review and any attached files.

Reviewer #1: No

Reviewer #2: No

---

## [Author Response · Author response to Decision Letter 0]

3 Dec 2022

Revision Overview: General comments for editorial board and reviewers

We previously submitted our manuscript entitled: “Self-organized metabotyping of obese individuals identifies clusters responding differently to bariatric surgery” to be considered for publication in PLOS ONE. In our revision, we carefully considered the comments regarding the confounding effects of age and gender, multiple hypothesis testing and the quality (resolution) of our Figures. As suggested by the editorial board, we also updated our Cover Letter to include potential changes in our Funding Sources and Data Availability. Herewith, we submit a point-by point response to the reviewers that outlines how we have addressed their concerns and thereby pursuing further consideration of this work at PLOS ONE.

Response to Reviewers

Reviewer #1:

This study performs a distinctive analysis in understanding the response in obesity to bariatric surgery, integrating gut microbiomics and plasma metabolome via the usage of SOMs, and further informing these profiles with tissue transcriptomics. Whilst this multi-omics strategy is conducted on a relatively small cohort of 106, the statistics are rigorous and provide a uniquely informed disease stratification.

1. Table 1 – please include a column listing demographics without being separated into the SOMs. This will help to understand the distribution of the cohort amongst the SOMs.

As suggested, a new column describing the BARIA population demographics has been added to Table 1, in order to make the overall distribution of the BARIA cohort more comprehensible:

Clinical Metadata SOM Cluster 1 SOM Cluster 2 SOM Cluster 3 SOM Cluster 4 SOM Cluster 5 BARIA

population

Demographic 

Participants (%) 17(16%) 29(27.4%) 25(23.6%) 18(17%) 17(16%) 106(100%)

Female (% Total Participants, % of SOM Cluster) 13(12.2%, 76.5%) 25(23.6%, 86.2%) 18(17%, 72%) 14(13.2%, 77.8%) 14(13.2%, 82.4%) 84(79.2%)

Male (% Total Participants, % of SOM Cluster) 4 (3.8%, 23.5%) 4 (3.8%, 13.78%) 7 (6.6%, 28%) 4 (3.8%, 22.22%) 3 (2.8%, 17.6%) 22(20.8%)

Anthropometric 

Age (years) 48(29-60)* 40(20-57)* 53(26-64)* 56(39-64)* 44(22-62)* 46(20-14)

BMI (kg/m2) 39.5(34-45.4) 38.2(32.9-60.9) 39.8(33-57.5) 38.3(33.8-47.1) 39.8(34.7-46.4) 39.42(32.9-70)

Waist circumference (cm) 125.3 ± 12.6 122.6 ± 12.3 123.7 ± 11.5 125.8 ± 12.2 123 ± 9.9 84.3 ± 57.7

Upper thigh circumference (cm) 135(120-149) 133(116-147) 130(103-165) 133(115-139) 136(123-144) 

122.5(103-165)

Total Body Fat (%) 53.6(41.6-64.7) 54.1(31.7-94.9) 51.8(39.3-104.8) 56.5(40.6-78.9) 57.6(44-64.5) 51(31.7-104.8)

Fat Free Mass (kg) 60.9(54.1-93.8) 59.6(50.3-90.6) 59.1(47.5-90.2) 59.8(49.5-85.1) 60.8(54-83.5) 58.9(47.5-93.8)

Systolic blood Pressure (mmHg) 131.5(116-156) 132(102-155) 133(108-161) 136(115-193) 135(115-157) 

132.5(102-193)

Diastolic blood Pressure (mmHg) 84.5(59-91) 81(54-99) 82(67-105) 80(45-121) 82(65-94) 

81(45-121)

Clinical lab values 

Fasting glucose (mmol/l) 5.8(4.8-11.4) 5.9(4.6-14.8) 5.7(5-13.8) 5.8(4.6-6.8) 5.6(4.5-8.7) 

5.8(4.5-14.8)

HbA1c (mmol/mol) 5.7(5.3-9.1) 5.7(4.6-9.8) 5.6(5-9.3) 5.8(5.2-6.9) 5.5(5.2-8.3) 5.7(4.6-9.8)

HOMA-IR 1.7(0.6-3.4) 1.6(0.5-6.9) 2.2(0.5-4.7) 1.3(0.8-4.8) 1.5(0.8-4.8) 1.6(0.6-6.9)

HOMA2-β 108.7(38.3-183.2) 87.9(29.1-227.8) 112(52.7-226.2) 92.1(52.4-357.8) 104.2(50.8-185.5) 93.5(29.1-357.8)

Total Cholesterol (mmol/l) 5.4 ± 1.1 4.6 ± 1 4.9 ± 1.1 5.3 ± 1.2 4.3 ± 0.9 4.9 ± 1.1

Triglycerides (mmol/l) 1.5(0.8-3.5) 1.3(0.6-5.8) 1.4(0.8-6) 1.4(0.8-5.9) 1.2(0.6-1.9) 1.4(0.6-6)

HDL Cholesterol (mmol/l) 1.2(0.8-1.8)* 1.1(0.6-1.9)* 1.1(0.7-2.5)* 1.2(0.7-2.1)* 1.2(1-2.7)* 

1.6(0.2-2.7)

LDL Cholesterol (mmol/l) 3.6 ± 1.1 2.9 ± 0.9 3.6 ± 0.9 3.4 ± 1.7 2.6 ± 0.8 3 ±1.1

Creatinine (�mol/l) 68(55-96) 63(46-83) 66(47-112) 75(56-172) 65(58-99) 

66(46-172)

Glomerular Filtration Rate (kl/1.73m2) 85(70-91)* 90(71-91)* 86(62-91)* 78(26-90)* 89(66-91)* 

88.5(26-91)

Baseline characteristics of the 106 BARIA participants included in the study. Data is expressed as mean ± standard deviation. Categorical variables are presented as numbers and percentages. Non-normally distributed variables are presented as median with interquartile range. For comparison among metabotypes Kruskal-Wallis test (extended Mann-Whitney U-test for multiple groups) was used. ‘*’ denotes differentially significant variables among the five metabotypes clusters (P<0.05). BMI: Body Mass Index, HbA1c: Hemoglobin A1c, HOMA-IR: Homeostatic Model Assessment of Insulin Resistance, HOMA-β: Homeostatic Model Assessment of beta-cell function, LDL: Low-Density Lipoprotein, HDL: High-Density Lipoprotein.

2. Line 148- 151: “652 metabolites were fully detected across all samples, 640 metabolites were partially detected across all samples, and 53 metabolites were not detected or failed to reach detection limit and therefore had a missing value”. Please clarify on the limit of detection here; the phrasing can be misinterpreted. What is the mean CV for those metabolites partially detected? Additionally, please clarify if these are relative or absolute abundances.

When METABOLON conducted the analysis of the BARIA metabolomics samples, it provided us raw, not normalized and unimputed absolute metabolite intensities. That means that for all the metabolites that could be detected, some of them had missing values. When dealing with LC-MS data, the reason for a missing value is not only its presence or absence within a sample. In Mass Spectrometry, when there is a particularly low abundance of a molecule, that might fail to be detected by the spectrometer. This does not entail that these molecules do not exist at all within the sample. Instead, they need to be more abundant in order to be detected by the spectrometer. The detection levels vary for each molecule and there is no universal standard for the minimum number of molecules that are required to be present in an injection mix, so as to detect the compound. Regarding the partially detected metabolites in the BARIA cohort, the mean abundance was 1 817 049, whereas the mean abundance for fully detected metabolites was 52 583 199.

3. Line 483: “dominated by Bacteroidetes and Firmicutes” – these genera are often increased in abundance in ageing. Given the acute age distribution of the cohort, have the authors considered this as confounding in this study of obesity? Line 566, states that “metabotype 2 represented the youngest individuals, yet the individuals being most heavily medicated for comorbidities…. In the gut microbiome Prevotella, Bacteroides and Lactobacillus species were found to be highly abundant”. Could these metagenome associated metabotypes be driven solely by age and sex? Given the small sample size, this is likely to have an impact on modelling. Please adjust models to correct for age and sex, and run training models to investigate the impact of age and sex on observed phenotypes.

As noted by Reviewer1, the abundance of some genera might be affected by ageing. We have therefore additionally adjusted for confounders in the metagenomics downstream computational analysis (age, gender). Confounder adjustment was conducted with the built-in function of DESeq2(1) package. The results can be seen below:

Review Fig 1: Differentially abundant metabolites and metabolic pathways among the five defined SOM clusters (metabotypes) adjusted for the confounding factors of age and gender. (A) Relative abundance and distribution of differentially significant metabolites among SOM and k-means defined clusters. Clusters two and three are most abundant in lipids (especially lysophospholipids and sphingomyelins) and amino acids (urea, arginine and proline metabolism). (B) Distribution of differentially significant metabolic pathways among SOM and k-means defined clusters, where numbers within each dot indicate how many metabolites of that particular specific pathway were differentially abundant across clusters. (C) Top 20 differentially significant metabolites among the SOM and k-means defined clusters, (P<0.05).

Removing these confounders (age and gender) did not affect the overall profiling of the dataset, especially in the most dominant Phyla of Bacteroidetes and Firmicutes. After adjustment, we observed 109 gut microbial species being differentially significant among SOM-defined clusters, instead of 288 that we had before. There might be an effect from age and gender in species’ levels, since the ones that initially had lower abundance have not been included in this dataset, such as different genera of Actinobacteria and Proteobacteria. Prevotella, Bacteroides and Lactobacillus remained the most dominant species with exactly the same distribution among the SOM-defined metabotypes. This finding reveals that the overall effect of the metabotyping is still present in the gut microbial metagenome even after regressing out confounders like age and gender. Review Fig 1 is now part of the supplementary files (S13 Fig).

4. Line 628- “liver genes enriched in oxidative phosphorylation”. Please correct to ‘hepatic function associated genes’.

Rephrased in the respective part of the manuscript, according to Reviewer1’s comment.

5. Line 640- “It is noteworthy that there are no distinct statistically significant responses in the weight loss or waist circumference reduction immediately after bariatric surgery (3 months after surgery), contrary to what would be expected”. In the methods the authors have listed clinical lab biochemistry measures, focusing more so on glucose and triglycerides. This would be important in understanding the clinical drivers of the metabotypes. It is stated, “pronounced changes in lysophospholipids, phosphatidylcholines, dicarboxylate fatty acids, sphingomyelins, and branched-chain amino acid metabolites among the five different metabotypes” – all of these metabolites are all implicated in inflammation. Have the authors looked at inflammatory markers such as CRP, and if that is lacking looking at red cell distribution width (RDW) as a proxy for inflammation?

The detected changes among metabotypes in lysophospholipids, phosphatidylcholines, dicarboxylate fatty acids, sphingomyelins, and branched-chain amino acid metabolites could be direct indicators of inflammation. A verification of that coming from direct clinical inflammatory markers, such as CRP would be very valuable. However, CRP at baseline did not exhibit any overall statistical significance among the SOM defined clusters as can be seen in Review Fig 2 below: 

Review Fig 2: C-Reactive protein associated with obesity and its statistical significance across the metabotypes (SOM & k-means defined clusters. Statistical significance among metabotypes is calculated with Kruskal-Wallis test and p valuehas been adjusted with FDR.

Only when conducting pairwise comparison among SOM-defined clusters we noticed a p=0.039, where p <0.05 for Clusters 2 and 5. Unfortunately, Red Cell Distribution Width, as an additional proxy for inflammation was not available and neither CRP or RDCW data where available at the 1-year follow up. Review Fig 2 is now part of the supplementary files (S15 Fig).

Reviewer #2:

This is a well-written manuscript using omics data to predict long-term weight loss. The authors used plasma metabolomics, adipose tissue gene expression data, and fecal metagenomics from individuals underwent bariatric surgery. The authors identified distinct metabotypes associated with immune function, fatty acid metabolism, protein signaling and obesity pathogenesis. The gut metagenome data identified metabotype associated with cardiometabolic comorbidities. However, several points need to be addressed by authors before considering for publication.

1. The manuscript is not clear whether you are using multi-omics or multiple types of omics data. The methods and results sections need to be re-written to clearly discuss this concept and the results associated.

Reviewer 2 has interestingly pinpointed the potential confusion that can be created with the terms “multiple omics” and “multi-omics”. Indeed, the manuscript utilises multiple types of omics data, such as metabolome, transcriptome and gut microbial metagenome. These datasets are in steps analysed separately in our computational pipeline, hence the term “multiple types of omics”. However, in order to extract a minimal set of a universal, discriminatory signature differentiating the metabotypes we have employed a multi-omics method, DIABLO. Understandably, there is a fine line distinguishing these two terms and we have rephrased to “multiple omics datasets” when referring to the metabolome, transcriptome and gut microbial metagenome and “multi-omics” when referring to the DIABLO multiple omics type integration.

2. Figure 1b, some clusters were associated with gender, and other characters not related to weight loss. Please regress out by taking the residuals confounders from the omics matrix.

There is a valuable concern regarding confounders in our computational approach and suggested framework. The point of metabotyping via SOMs is that it is a purely data-driven approach, not dependent by clinical labels. The metabolome acts as a direct read-out of all the metabolic activities across different tissues and host-microbiota interactions. It is therefore crucial to include as much data variability as possible and to limit external bias parameters when attempting to identify underlying metabolic phenotypes, especially when employing unsupervised machine learning algorithms.

If the corrections for covariates and confounding factors take place prior to the SOM method, it entails that we ourselves introduce bias to the dataset. The ANN SOM method is completely impartial to the original dataset and its’ metadata. The SOM method aims to cluster the raw metabolomics into sub-types and then we associate clinical variables to metabotypes. As a result, this type of analysis is not suitable for de-confounding. Since the SOM method only calculates distances, it cannot account for any confounding factors, hence all the metadata are introduced in a posterior phase. When we conduct the differential significance analysis in each omics dataset, we can actually account for confounders, so it is at this stage where we can see if indeed there is an effect. Hence, the interpretability of the results is affected by covariates and not the actual extraction of the metabotypes.

According to Reviewer’s 2 suggestion, we have now regressed out potential confounding factors from each omics matrix, such as age and gender, and have expanded our Results and Discussion sections in the revised manuscript accordingly. For the metabolomics dataset, after adjusting for age and gender, via the built-in function of HybridMTest(2), we obtained the following figure:

Review Fig 3. Differentially abundant metabolites and metabolic pathways among the five defined SOM clusters (metabotypes), adjusted for the confounding factors of age and gender. (A) Relative abundance and distribution of differentially significant metabolites among SOM and k-means defined clusters. Clusters two and three are most abundant in lipids (especially lysophospholipids) and amino acids (urea, arginine and proline metabolism). (B) Distribution of differentially significant metabolic pathways among SOM and k-means defined clusters, where numbers within each dot indicate how many metabolites of that particular specific pathway were differentially abundant across clusters. (C) Top 20 differentially significant metabolites among the SOM and k-means defined clusters, (P<0.05).

Clearly when we compare Fig 2 from the original manuscript to Review Fig 3, we can see that there is a difference in the number of the differential significant metabolites. We now observed 174 metabolites being differentially significant among SOM-defined clusters, instead of 226 that we had before. In spite of that, the relative distribution of differentially significant pathways and their % pathway abundance remains identical, as seen in Review Fig 3A. As far as the Metabolite Specific Pathways is concerned, the same pathways appear to be most abundant, such as lysophospholipids, phosphatidylcholines, dicarboxylate fatty acids, and branched-chain amino acid metabolites and also maintain their distribution among the metabotypes. What has changed, is that metabolite compounds that were less abundant in the initial dataset have now been eliminated after confounder adjustment. This Figure is now part of the supplementary files (S11 Fig).

After accounting for age and gender in our DeSEq2 models, in the differential significance analysis of the transcriptome, we could indeed notice a more pronounced difference, as seen below in Review Fig 4:

Review Fig 4. Differentially enriched KEGG metabolic pathways among the five defined SOM clusters (metabotypes)), adjusted for the confounding factors of age and gender. (A) Top 15 differentially enriched KEGG metabolic pathways for hepatic transcriptome among the SOM and k-means defined clusters, ranked based on their scores after differential gene expression analysis (DESeq2, P<0.05) and gene set analysis (GSA with EnrichR). (B) Top 15 differentially enriched KEGG metabolic pathways for mesenteric adipose transcriptome among the SOM and k-means defined clusters, ranked based on their scores after differential gene expression analysis (DESeq2, P<0.05) and gene set analysis (GSA) with EnrichR). (C) Top 10 differentially enriched KEGG metabolic pathways for subcutaneous adipose tissue transcriptome among the SOM and k-means defined clusters, ranked based on their scores after differential gene expression analysis (DESeq2, P<0.05) and gene set analysis (GSA with EnrichR). (D) 20 highest scoring KEGG metabolic pathways according to EnrichR GSA score for liver, mesenteric adipose and subcutaneous adipose tissues. Z score indicates different levels of differentially expressed pathways, for each SOM and k-means defined cluster.

This is expected since even if the genetic making(genome) of men and women is the same, the transcriptome is distinctly dimorphic with dissimilar disease susceptibilities(3). Most of the changes were on the liver transcriptome. As in our initial analysis the metabolic pathways enriched within the hepatic transcriptome exhibited mixed directionality in regulation, but this time pathogenic pathways along with compound degradation pathways were more dominant, especially in clusters 1 and 2. In general, after the confounders removal there is a more distinct pattern of nutrients catabolism, degradation and absorption and less presence of inflammatory pathways in the hepatic transcriptome when comparing to the pathways that we obtained in our previous enrichment analysis. Surprisingly, the mesenteric adipose tissue was again enriched for amino acid metabolic processes, in fact the same pathways as before including phenylalanine, tyrosine, tryptophan biosynthesis and fatty acids metabolism. Amino acid metabolic pathways in mesenteric adipose tissue exhibited consistent upregulation in clusters 4 and 5, as prior to the age and gender adjustment. In subcutaneous adipose tissue we also obtained a different set of enriched metabolic pathways, this time not including pathogenic pathways. Instead, we observed similar pathways as in the mesenteric adipose tissue, primarily amino acid and fatty acid metabolic pathways. Review Fig 4 is now part of the supplementary files (S12 Fig).

All in all, we can still identify the same or complementary pathways that distinguish the SOM-defined clusters, even after the confounders’ correction. Once more, transcriptome analysis from these three tissues showed distinct regulation of lipid, amino acid, and pathogenic pathways amongst the metabotypes.

We have also adjusted the gut microbial metagenome for age and gender. The results are shown in Review Fig 2 (Fig S15) above.

As mentioned above in our response to Reviewer 1 Comment 3, some species indeed differ, especially the ones that were lowly abundant, such as Actinobacteria and Proteobacteria. However, similar to what occurred in the metabolomic dataset adjustment the distribution among metabotypes did not alter after accounting for age and gender. 

3. Figure 5a, 5b, 5c, 5d, 5e, it seems the clusters didn't separate out well long-term weight loss, please provide statistics.

As noted by Reviewer 2, not all clusters have separated weight loss in a way that yielded statistically significant results. The reasons why not all statistical significance results were annotated on the figure on the original manuscript were the facilitation of the image’s legibility and the provision of straightforward significant results to the readership. T-test was used to evaluate the statistical significance among the metabotypes and the results were subjected to multiple hypothesis testing and adjustment with FDR (5%). In the following updated version of Fig 5, all the statistics are provided in detail: 

Review Fig 5. Weight and fat loss progression at distinct time points after bariatric surgery for the five defined SOM clusters (metabotypes). (A) Weight (kg) of BARIA individuals at baseline, three months, six months and one year after bariatric surgery for each metabotype. (B) Weight loss(kg) of BARIA individuals at baseline, three months, six months and one year after bariatric surgery for each metabotype. (C) Waist circumference (cm) of BARIA individuals at baseline, three months, six months and one year after bariatric surgery for each metabotype. (D) Reduction of waist circumference(cm) of BARIA individuals at baseline, three months, six months and one year after bariatric surgery for each metabotype. (E) Upper leg circumference (cm) of BARIA individuals at baseline, three months, six months and one year after bariatric surgery for each metabotype. (F) Reduction of upper leg circumference(cm)of BARIA individuals at baseline, three months, six months and one year after bariatric surgery for each metabotype. Statistical significance among metabotypes is calculated with t-test and adjusted with FDR; the symbols indicating significance among metabotypes are ‘*’: P<=0.05

4. This data leveraged many features, and no discussion of multiple testing / ways to control for multiple testing. If it is based on nominal significance, please state in the abstract, results, and discussion.

Thank you for this comment. All our results have been through multiple hypothesis testing and our findings are not based on nominal significance. For each dataset, the multiple hypothesis testing method and respective adjustment chosen for each omic dataset has been described in the Methods section. For metabolomics we have employed ANOVA and Kruskal Wallis tests (given the different distributions of the metabolites), with the use of HybridMTest package(2). For the metabolites that were significantly differential with both ANOVA and Kruskal Wallis methods, HybridMTest package performed hybrid multiple hypothesis testing using the empirical Bayes probability and adjusted for the results with this method accordingly. DESeq2(1) package in R (version 3.6.3) was used for the transcriptome and gut microbial metagenome differential significance analysis .The statistical analysis method for calculating differential expression rates was the LRT test (log-ratio test). Naturally, the results were subjected to multiple hypothesis testing and FDR correction (FDR 5%), with the DESeq2 built-in IHW(4) package.

5. Again, many of the results discussed clusters that reflect gender, age and etc. Please take out variations of other confounders by regressing out these variables.

We have now regressed out potential confounding factors, such as age and gender, as suggested by both reviewers and have expanded our results and discussion sections accordingly. Please read our response above, in Comment 2.

6. The resolution of the figures are not good, please modify.

We have now modified the resolution of the requested figures, from 300dpi to 600 dpi, according to the comments by Reviewer2 and the Editorial Team.

Academic Editor

1. Please ensure that your manuscript meets PLOS ONE's style requirements, including those for file naming. The PLOS ONE style templates can be found at: https://journals.plos.org/plosone/s/file?id=wjVg/PLOSOne_formatting_sample_main_body.pdf and :https://journals.plos.org/plosone/s/file?id=ba62/PLOSOne_formatting_sample_title_authors_affiliations.pdf

We have carefully examined the PLOS ONE’s style requirements, as provided in the links above and have integrated those in our revised manuscript.

"The BARIA study is funded by the Novo Nordisk Foundation (NNF15OC0016798). The Novo Nordisk Foundation Center for Basic Metabolic Research is supported by an unconditional grant (NNF10CC1016515) from the Novo Nordisk Foundation to University of Copenhagen. The BARIA study is a Scandinavian-Dutch collaboration. Funding from Knut and Alice Wallenberg Foundation is also acknowledged."

In our revised Cover Letter, we updated our financial disclosure in order to include the statement: “There was no additional external funding received for this study.”

"The computations and RNA Sequencing were enabled by resources provided by the Swedish National Infrastructure for Computing (SNIC) at C3SE (SNIC Computational Center of Chalmers University of Technology) partially funded by the Swedish Research Council through grant agreement no. 2018-05973."

"The BARIA study is funded by the Novo Nordisk Foundation (NNF15OC0016798). The Novo Nordisk Foundation Center for Basic Metabolic Research is supported by an unconditional grant (NNF10CC1016515) from the Novo Nordisk Foundation to University of Copenhagen. The BARIA study is a Scandinavian-Dutch collaboration. Funding from Knut and Alice Wallenberg Foundation is also acknowledged."

The statement “The computations and RNA Sequencing were enabled by resources provided by the Swedish National Infrastructure for Computing (SNIC) at C3SE (SNIC Computational Center of Chalmers University of Technology) partially funded by the Swedish Research Council through grant agreement no. 2018-05973” has now been relocated from the Acknowledgements and appended to the updated Funding Statement in our revised Cover Letter.

Any funding related Text is now removed from the Revised Manuscript, while we have appended the amendment for the Funding Statement in our revised Cover Letter, as suggested by the Academic Editor

"The BARIA study is funded by the Novo Nordisk Foundation (NNF15OC0016798). The Novo Nordisk Foundation Center for Basic Metabolic Research is supported by an unconditional grant (NNF10CC1016515) from the Novo Nordisk Foundation to University of Copenhagen. The BARIA study is a Scandinavian-Dutch collaboration. Funding from Knut and Alice Wallenberg Foundation is also acknowledged."

Please state what role the funders took in the study. If the funders had no role, please state: ""The funders had no role in study design, data collection and analysis, decision to publish, or preparation of the manuscript."" If this statement is not correct you must amend it as needed. 

In our revised Cover Letter, we updated our financial disclosure with the section Role of Funder in order to include the statement: “The funders had no role in study design, data collection and analysis, decision to publish, or preparation of the manuscript”

The amended Data Availability Statement, that addresses ethical or legal concerns and includes contact information for data access committees, is now included both in our revised Cover Letter and our Revised Manuscript.

All the Changes in our Data Availability Statement are now part of our revised Cover Letter and the Revised Manuscript.

7. Your ethics statement should only appear in the Methods section of your manuscript. If your ethics statement is written in any section besides the Methods, please move it to the Methods section and delete it from any other section. Please ensure that your ethics statement is included in your manuscript, as the ethics statement entered into the online submission form will not be published alongside your manuscript. 

Our Ethics Statement is part of the Methods section in the Revised Manuscript.

8. Please upload a new copy of Figure 2 and Supporting Figures 5, 8, 9, 10 and 11 as the detail is not clear. Please follow the link for more information: https://blogs.plos.org/plos/2019/06/looking-good-tips-for-creating-your-plos-figures-graphics/

https://blogs.plos.org/plos/2019/06/looking-good-tips-for-creating-your-plos-figures-graphics/

Updated copies of :

• Fig 2 as Fig2_Review_TiFF_600dpi

• S5 Fig as S5_Fig_TiFF_600dpi

• S8 Fig as S8_Fig_TiFF_600dpi

• S9 Fig as S9_Fig_TiFF_600dpi

• S10 Fig as S10_Fig_TiFF_600dpi

• S11 Fig as S14_Fig_TiFF_600dpi Fig, since we added four additional supplementary figures in our submission.

With higher resolution are uploaded in in the submission platform along with other revision materials.

After the Academic Editor’s suggestion, we went through our Reference List in order to remove and replace potentially retracted articles and consequently updated our citations.

References:

1. Love MI, Huber W, Anders S. Moderated estimation of fold change and dispersion for RNA-seq data with DESeq2. Genome Biol. 2014 Dec;15(15):550. 

2. Stan Pounds DF. HybridMTest: Hybrid Multiple Testing. 2019. 

3. Gershoni M, Pietrokovski S. The landscape of sex-differential transcriptome and its consequent selection in human adults. BMC Biol. 2017;15(1):1–15. 

4. Ignatiadis N, Klaus B, Zaugg JB, Huber W. Data-driven hypothesis weighting increases detection power in genome- scale multiple testing. 2016;13(7).

---

## [Editor Report · Decision Letter 1]

6 Dec 2022

Self-organized metabotyping of obese individuals identifies clusters responding differently to bariatric surgery

PONE-D-22-23260R1

Dear Dr. Nielsen,

We’re pleased to inform you that your manuscript has been judged scientifically suitable for publication and will be formally accepted for publication once it meets all outstanding technical requirements.

Kind regards,

Priyadarshini Kachroo

Academic Editor

PLOS ONE
---

## [Editor Report · Acceptance letter]

19 Dec 2022

PONE-D-22-23260R1 

Self-organized metabotyping of obese individuals identifies clusters responding differently to bariatric surgery 

Dear Dr. Nielsen:

I'm pleased to inform you that your manuscript has been deemed suitable for publication in PLOS ONE. Congratulations! Your manuscript is now with our production department. 

Kind regards, 

on behalf of

Dr. Priyadarshini Kachroo 

Academic Editor

PLOS ONE